# Interleukin-12 bypasses common gamma-chain signalling in emergency natural killer cell lymphopoiesis

Isabel Ohs[1], Maries van den Broek[2], Kathrin Nussbaum[1], Christian Münz[3], Sebastian J. Arnold[4,5], Sergio A. Quezada[6], Sonia Tugues[1,*] & Burkhard Becher[1,*]

Differentiation and homeostasis of natural killer (NK) cells relies on common gamma-chain ($\gamma$c)-dependent cytokines, in particular IL-15. Consequently, NK cells do not develop in mice with targeted $\gamma$c deletion. Herein we identify an alternative pathway of NK-cell development driven by the proinflammatory cytokine IL-12, which can occur independently of $\gamma$c-signalling. In response to viral infection or upon exogenous administration, IL-12 is sufficient to elicit the emergence of a population of CD122$^+$CD49b$^+$ cells by targeting NK-cell precursors (NKPs) in the bone marrow (BM). We confirm the NK-cell identity of these cells by transcriptome-wide analyses and their ability to eliminate tumour cells. Rather than using the conventional pathway of NK-cell development, IL-12-driven CD122$^+$CD49b$^+$ cells remain confined to a NK1.1$^{low}$NKp46$^{low}$ stage, but differentiate into NK1.1$^+$NKp46$^+$ cells in the presence of $\gamma$c-cytokines. Our data reveal an IL-12-driven hard-wired pathway of emergency NK-cell lymphopoiesis bypassing steady-state $\gamma$c-signalling.

[1] Inflammation Research, Institute of Experimental Immunology, University of Zurich, 8057 Zurich, Switzerland. [2] Tumor Immunology, Institute of Experimental Immunology, University of Zurich, 8057 Zurich, Switzerland. [3] Viral Immunobiology, Institute of Experimental Immunology, University of Zurich, 8057 Zurich, Switzerland. [4] Institute of Experimental and Clinical Pharmacology and Toxicology, Faculty of Medicine, and BIOSS Centre of Biological Signalling Studies, Albert-Ludwigs-University, D-79104 Freiburg, Germany. [5] BIOSS Centre of Biological Signalling Studies, Albert-Ludwigs-University, D-79104 Freiburg, Germany. [6] Cancer Immunology Unit, Research Department of Hematology, University College London Cancer Institute, WC1E 6BT London, UK. * These authors contributed equally to this work. Correspondence and requests for materials should be addressed to B.B. (email: becher@immunology.uzh.ch).

As main components of the innate immune system, NK cells play a key role in controlling infections and limiting cancer progression[1,2]. Recognition of infected or transformed cells by NK cells involves a plethora of activating and inhibitory receptors, that in combination determine whether a target cell will be killed or spared[3]. The elimination of target cells is achieved via death receptor pathways or the release of cytotoxic granules containing perforin and granzyme[4,5].

In addition to their cytotoxic function, NK cells are a major source of proinflammatory cytokines such as tumour necrosis factor alpha (TNF-α) and interferon gamma (IFN-γ), which activate the myeloid compartment to join the fight against infections or cancer[6]. In turn, cytokines can modulate NK-cell responses[7]. More specifically, interleukin (IL)-15, which together with other cytokines (IL-2, IL-4, IL-7, IL-9 and IL-21) signals through the γc subunit, is critical for NK-cell development, homeostasis and activation[8]. Once lineage committed, as seen by acquiring IL-2/15Rβ (CD122) expression, NK cells require continuous IL-15R engagement for further differentiation and maintenance[9,10]. Accordingly, mice deficient in IL-15, IL-15Rα or γc are devoid of NK cells[11,12]. One study reported an expansion of lymphocytes with an NK-cell phenotype in $Rag2^{-/-}Il2rg^{-/-}$ mice upon murine cytomegalovirus (MCMV) infection, suggesting that NK cells can expand in the absence of IL-15 under highly inflammatory conditions[13]. Even though the cytokine IL-12 was involved in this process, whether the expanded population represents a unique IL-15-independent NK-cell subset remains elusive.

IL-12 is secreted by myeloid cells during inflammation, and its primary effects on NK cells include increased IFN-γ production, proliferation and the expression of cytotoxic mediators[14–20]. So far, the effect of IL-12 on NK cells has been reported for mature NK (mNK) cells, but it is currently unknown whether IL-12 contributes to NK-cell development in the BM.

Here we discovered that IL-12 drives a pathway of emergency NK-cell lymphopoiesis independently of γc-signalling. IL-12 initiates the generation of a cell population expressing CD49b and CD122, but low levels of the lineage markers NK1.1 and NKp46. Using a comprehensive phenotypic and functional characterization, we provide evidence that these cells represent a population of NK cells with a unique cell-surface receptor repertoire resembling immature NK cells. Mechanistically, IL-12-driven CD122$^+$CD49b$^+$ NK cells differentiate from NKPs in the BM through an unconventional developmental pathway. We demonstrated the emergence of IL-12-induced NK cells during viral infection as well their ability to clear tumour cells and limit metastatic spread, reinforcing the relevance of this alternative process of NK-cell lymphopoiesis.

## Results

**IL-12 induces γc-independent emergency NK-cell lymphopoiesis.** NK cells were reported to expand independently of γc-signalling in response to MCMV infection[13], a hitherto unrecognized process whose underlying mechanism remains ill-defined. To investigate whether the expansion of such NK cells only occurs during MCMV infection or is a general response to inflammation, we infected wild-type (WT) and $Rag2^{-/-}Il2rg^{-/-}$ mice (lacking T, NKT and B cells as well as ILCs) with vaccinia virus (VV), a poxvirus controlled by NK cells early after infection[21]. In lungs of $Rag2^{-/-}Il2rg^{-/-}$ C57BL/6 mice, VV infection induced the expansion of a population of lymphocytes expressing both CD122 and CD49b (Fig. 1a), two markers that phenotypically define NK cells[22]. Viral infections trigger the release of inflammatory mediators such as IL-12 (ref. 23), and we detected elevated amounts of IL-12 in the serum of VV-infected

WT as well as $Rag2^{-/-}Il2rg^{-/-}$ mice (Fig. 1b). Moreover, neutralization of IL-12 prevented the expansion of CD122$^+$CD49b$^+$ cells in infected mice (both WT and $Rag2^{-/-}Il2rg^{-/-}$; Fig. 1c), indicating a dependence of this population on IL-12. Altogether, these results highlight IL-12 as a key cytokine to initiate the emergence of γc-signalling-independent CD122$^+$CD49b$^+$ cells (hereafter called emergency NK (eNK) cells) upon viral infection.

**IL-12 generates unconventional but functional NK cells.** We next addressed whether IL-12 promotes the development of eNK cells in the absence of infection. Indeed, we found eNK cells, which morphologically resembled lymphocytes, in lungs of $Rag2^{-/-}Il2rg^{-/-}$ mice treated with IL-12 (Fig. 2a,b). The population of eNK cells was shown to express low levels of NK1.1 (Fig. 2a). Of note, a subset of CD122$^+$CD49b$^+$ cells with low NK1.1 expression was also found in lungs of IL-12-treated WT mice (Fig. 2a), indicating that the IL-12-induced expansion of these cells also occurs in the face of physiological γc-signalling. eNK cells expressed markers typically associated with conventional NK (cNK) cells such as the transcription factor Eomesodermin (Eomes) and the activating receptor NKG2D (Fig. 2c). In comparison with the NK1.1$^+$ subset of CD122$^+$CD49b$^+$ cells from control (NK) or IL-12-treated WT mice (NK + IL-12), eNK cells expressed low amounts of NKp46 (Fig. 2c) and unusually high levels of DNAM-1 (Fig. 2c), as well as a differentiated (CD11b$^{high}$CD27$^{high}$KLRG1$^+$) phenotype (Supplementary Fig. 1a). Also, eNK cells did not express the IL-7 receptor (CD127; Supplementary Fig. 1b) and exhibited a high proliferation rate accompanied by reduced levels of Bcl-2 (Fig. 2d; Supplementary Fig. 1c), an anti-apoptotic factor involved in peripheral NK-cell survival by IL-15 (ref. 10). Importantly, NK1.1$^{low}$ NK + IL-12 cells from WT mice phenotypically closely resembled the population of highly proliferative, less differentiated Eomes$^+$ eNK cells (Fig. 2c,d; Supplementary Fig. 1a,c).

We next investigated the ability of eNK cells to produce IFN-γ, the main cytokine secreted by cNK cells upon IL-12 stimulation[20]. IFN-γ was secreted by lung CD122$^+$CD49b$^+$ NK1.1$^{low}$ cells not only in IL-12-treated $Rag2^{-/-}Il2rg^{-/-}$ mice, but also in WT mice, where this population contributed to almost 40% of the total IFN-γ production (Fig. 2e). Furthermore, the responses of IL-12-induced eNK cells to YAC-1 targets demonstrated their cytotoxic potential (Fig. 2f). Taken together, IL-12 bypasses the requirement of γc-signalling for NK-cell lymphopoiesis by inducing the generation of an unconventional population of eNK cells with cytotoxic properties.

**eNK cells represent a distinct NK-cell subset.** To further explore in which extent eNK cells mirror NK-cell features, we performed deep transcriptome analysis of sorted eNK cells as well as NK and NK + IL-12 cells from WT mice. Principal component (PC) analyses indicated that each group of cells formed individual clusters in the PC space (Fig. 3a). Basic hierarchical clustering of the three populations showed that eNK cells were more closely related to NK + IL-12 cells than they were to NK cells (Fig. 3b). Similar to NK cells, eNK cells expressed high amounts of Eomes, killer-cell lectin-like receptors (Klr), Fc receptors and adhesion molecules, thus confirming the NK-cell identity of this population (Fig. 3c; Supplementary Table 1). The transcripts differentially expressed in eNK compared with NK + IL-12 cells revealed, however, a gene signature reminiscent of immature NK cells, as shown by high expression of Cd27, Cd69, Cxcr3 and Cxcr6 (Fig. 3c; Supplementary Table 1). In contrast, transcripts mainly confined to mNK cells[24–28], such as the integrins CD49b (Itga2) and CD11b (Itgam), Cx3cr1, S1pr5 and several members of the Ly49 receptor family (Klra1, Klra3, Klra9 and Klra7), were

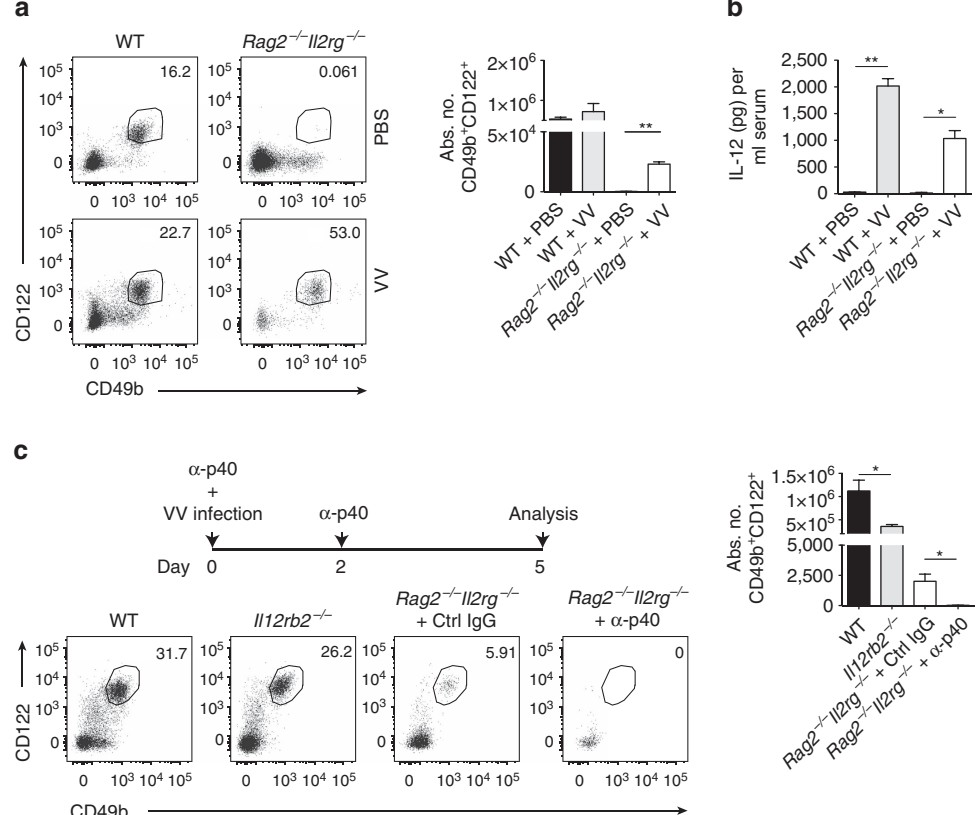

**Figure 1 | VV infection generates IL-12-dependent CD122+CD49b+ NK cells.** (**a**) Representative plots and quantification of lung CD45+CD3−CD122+CD49b+ cells of uninfected versus VV-infected WT or $Rag2^{-/-}Il2rg^{-/-}$ mice. Data represent two similar experiments ($n = 3–5$ mice per group per experiment). (**b**) IL-12 serum levels in VV-infected WT or $Rag2^{-/-}Il2rg^{-/-}$ mice were compared to controls. Data represent two independent experiments ($n = 3–5$ mice per group per experiment). (**c**) Schedule of α-p40 administration in VV-infected mice. Representative plots and quantification of lung CD45+CD3−CD122+CD49b+ cells of VV-infected WT, $Il12rb2^{-/-}$ or $Rag2^{-/-}Il2rg^{-/-}$ mice treated with Ctrl IgG or α-p40. Data represent two independent experiments with at least three mice per group. Statistical analysis was performed (*$P < 0.05$, **$P < 0.01$, unpaired Student's t-test without (**a**) or with (**b,c**) Welch's correction) and error bars represent ± s.e.m.

present albeit downregulated in eNK cells (Fig. 3c; Supplementary Table 1). The immature profile of eNK cells was not associated with decreased levels of *Eomes* and T-bet (*Tbx21*) (Fig. 3c; Supplementary Table 1), but with changes in cytokine production and cytotoxic effectors. Thus, whereas NK + IL-12 cells expressed higher amounts of granzyme (*Gzma* and *Gzmb*) and perforin (*Prf1*) compared to eNK cells, the latter upregulated GM-CSF (*Csf2*), lymphotoxin-β (*Ltb*), TNF-α (*Tnf*) and TNF-Related Apoptosis Inducing Ligand (TRAIL; *Tnfsf10*) (Fig. 3c; Supplementary Table 1) usually acquired at earlier stages of NK-cell maturation[29].

We examined the cell surface levels of several NK-cell receptors to further assess the maturation status of lung eNK cells. In general, whereas NK1.1+ NK + IL-12 cells were phenotypically similar to the NK-cell population from Ctrl-treated WT mice, the phenotype of eNK cells resembled that of NK1.1^low NK + IL-12 cells from WT mice (Fig. 3d). Thus, eNK cells and NK1.1^low NK + IL-12 cells from WT mice expressed high levels of CD94, NKG2A/C/E and TRAIL, but low amounts of Ly49D (*Klra4*) and Ly49I (*Klra9*) when compared with NK and NK1.1+ NK + IL-12 cells from WT mice (Fig. 3d). Low levels of Ly49G2 (*Klra7*) and Ly49A (*Klra1*) were also observed in eNK cells (Fig. 3d). Importantly, this was not exclusive of eNK cells in the lung microenvironment, since liver eNK cells showed a very similar overall cell surface phenotype (Supplementary Fig. 2a,b).

Altogether, these findings place eNK cells into a distinct NK-cell subset with immature characteristics. The resemblance

between NK1.1^low NK + IL-12 cells of WT mice and eNK cells reinforces the existence of an IL-12-induced pathway of alternative NK-cell lymphopoiesis also when γc cytokines are present.

**IL-12 initiates a different pathway of NK-cell development.** The unconventional phenotype observed in lung eNK cells raised the question of how these cells originate in the BM. NK cells develop via four stages from (i) lineage negative (Lin−) CD127+CD135+ common lymphoid progenitors (CLPs) into (ii) Lin−CD244+CD127+CD135− pre-NK-cell progenitors (pre-NKPs) and further into (iii) Lin−CD244+CD27+CD122+CD49b−NK1.1− NKPs, which acquire the expression of the IL-15R complex CD122 and ultimately give rise to mature (iv) CD244+CD122+CD49b+NK1.1+ NK cells[30,31]. We used unsupervised non-linear dimensionality reduction (t-SNE[32]) to identify and visualize these distinct NK-cell developmental stages in the BM of IL-12-treated WT and $Rag2^{-/-}Il2rg^{-/-}$ mice (Fig. 4a). With this approach, the CD45+Lin−CD122+ population could be subdivided into demarcated cluster sets defined by differential expression of several NK-cell markers (Fig. 4a–c). Thus, cluster 1, negative for all NK-cell markers but for CD27 and present in both WT and $Rag2^{-/-}Il2rg^{-/-}$ mice, harbours NKPs (Fig. 4a–c). The population of mNK cells was clearly identified by clusters 3 and 6 in WT mice, representing less (CD27^high CD11b^low) and more differentiated (CD27^low CD11b^high KLRG1+) mNK cells, respectively

(Fig. 4a–c). Both in WT and $Rag2^{-/-}Il2rg^{-/-}$ mice, IL-12 induced the emergence of eNK cells (DX5$^+$NK1.1$^-$NKp46$^{low}$ CD11b$^+$CD27$^{low}$KLRG1$^{high}$ cell in cluster 5), and an unexpected population of DX5$^{low}$NK1.1$^-$NKp46$^-$CD11b$^-$CD27$^-$ KLRG1$^-$ cells (cluster 4), probably representing an intermediate precursor stage of NK-cell development (Fig. 4a–c). Also in the BM, we observed the expression of Eomes in eNK cells by using an Eomes fluorescent reporter mouse (Supplementary Fig. 3a).

Moreover, when transferred into lymphopenic mice, IL-12-driven CD122$^+$CD49b$^+$NK1.1$^{low}$ eNK cells isolated from WT mice developed into NK1.1$^+$NKp46$^+$ cells, indicating that the presence of γc-cytokines is required for the acquisition of these NK-cell lineage markers (Fig. 4d; Supplementary Fig. 3b).

IL-12 expanded the populations of NKPs and eNK cells in the BM of both WT and $Rag2^{-/-}Il2rg^{-/-}$ mice, whereas CLPs and pre-NKPs in WT mice remained unaltered (Fig. 4e,f;

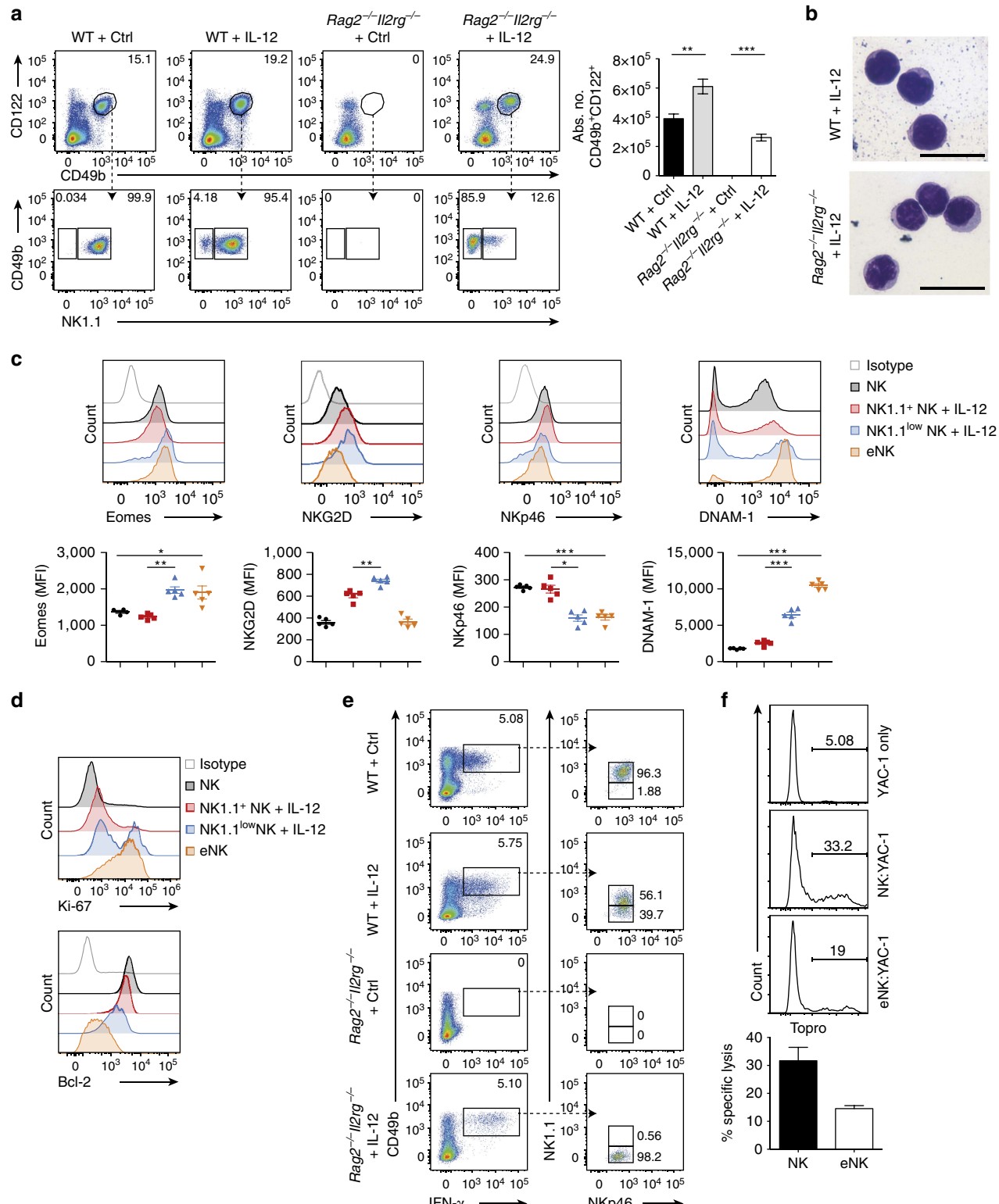

Supplementary Fig. 3c,d). Accordingly, proliferation of NKPs as well as of eNK and mNK cells was increased upon IL-12 treatment (Supplementary Fig. 3e). Overall, these results point towards an hitherto unrecognized pathway of NK-cell development regulated by IL-12, which bypasses canonical γc-chain signalling.

**NKPs respond to IL-12 and differentiate into eNK cells.** To determine the lymphoid precursor population of eNK cells, we sorted CLPs, pre-NKPs, NKPs, eNK and mNK cells from BM of WT mice and quantified *Il12rb2* transcripts. *Il12rb2* was expressed by NKPs, eNK and mNK cells, but not by CLPs and only at low levels by pre-NKPs (Fig. 5a). Furthermore, we found higher amounts of *Il12rb2* transcripts in NKPs and eNK cells from IL-12-treated compared with Ctrl-treated WT mice (Fig. 5a), indicating that IL-12 induces the expression of its own receptor complex in NKPs. Functional IL-12R engagement was further shown by the expression of *Tbx21* and *Ifng* transcripts, which occurred only in the three NK-cell populations that express *Il12rb2* (Fig. 5b,c).

To further assess direct effects of IL-12 on NKPs and eNK cells, we isolated these cell populations from the BM of WT mice and treated them for 6 h with IL-12 *in vitro*. *Ifng* expression was increased upon IL-12 stimulation in both NKPs and eNK cells, indicating a direct signal through the IL-12R in both cell types (Fig. 5d). IL-12 was sufficient to drive differentiation of NKPs into eNK cells within whole BM suspension or highly purified NKPs kept on monolayers of OP-9 stromal cells (Fig. 5e,f). Collectively, these data highlight a pivotal role of IL-12 in NK-cell differentiation by acting on NKPs, the stage at which IL-15 is required for the maturation of these cells during steady-state lymphopoiesis.

**eNK cells display anti-tumour activity.** The observed cytotoxic properties of eNK cells prompted us to test their role in tumour surveillance. First, we used TRAIL-sensitive MC38 tumour cells in C57BL/6 mice. Intravenously inoculated MC38-GFP cells were efficiently eliminated from lungs of IL-12-treated *Rag2*[−/−] *Il2rg*[−/−] mice, in comparison with the high numbers of tumour cells found in Ctrl-treated *Rag2*[−/−] *Il2rg*[−/−] mice (Fig. 6a). The dependence of this process on eNK cells was shown upon their depletion with an anti-asialo GM1 antibody, which completely abolished the clearance of MC38-GFP target cells (Fig. 6a).

We next tested the metastasis-controlling capacity of eNK cells in the orthotopic 4T1 model of breast cancer [33]. In this model, the inoculation of 4T1 cells in *Rag2*[−/−] *Il2rg*[−/−] mice resulted in an increased metastatic burden compared to WT and *Rag1*[−/−] mice (lacking T, NKT and B cells; Fig. 6b), thus indicating the importance of ILCs in limiting metastatic dissemination. IL-12

treatment of tumour-bearing WT mice led to a significant reduction of lung metastases (Fig. 6c). This was not due to a direct effect of IL-12 on 4T1 tumour cells, since these cells did not express *IL-12Rβ2* and *IL-12Rβ1* and their *in vitro* growth was not affected by IL-12 (Supplementary Fig. 4). Remarkably, IL-12 elicited tumour control also in tumour-bearing *Rag2*[−/−] *Il2rg*[−/−] mice (Fig. 6c), demonstrating that IL-12 induces protective immunity even in mice lacking all T, B and ILCs. In this setting, IL-12 treatment of 4T1 tumour-bearing *Rag2*[−/−] *Il2rg*[−/−] mice led to the expansion of eNK cells from the BM (Fig. 6d,e). These cells were identified as the main IL-12-responsive cell type, as they virtually produced all the IFN-γ detected in *Rag2*[−/−] *Il2rg*[−/−] mice (Fig. 6f). The dependence of tumour rejection on IFN-γ-producing eNK cells was solidified by the finding that IL-12-treated IFN-γ receptor deficient (*Ifngr*[−/−]) mice failed to control lung metastasis (Fig. 6g). Moreover, the depletion of eNK cells reversed IL-12-mediated suppression of metastasis in 4T1 tumour-bearing *Rag2*[−/−] *Il2rg*[−/−] mice (Fig. 6h). Collectively, the contribution of eNK cells to tumour control highlights the importance of γc-independent emergency NK-cell lymphopoiesis in tumour protective immunity.

## Discussion

Even though γc-cytokines regulate various aspects of NK-cell biology, only IL-15 is essential for NK-cell development[7,10]. Under physiological conditions, NK cells develop in the BM from lymphoid precursor cells[34], and this maturation pathway is aborted in the absence of IL-15R signalling[11]. Here, we demonstrated that IL-12 can bypass this developmental road block and induce unconventional eNK cells independently of γc-signalling. We propose that this pathway serves to combat infections or other threats to normal tissue homeostasis (for example, cancer) and can be exploited therapeutically.

Early evidence of IL-15-independent NK-cell expansion upon CMV infection[13] suggested that mammals might utilize alternative mechanisms to ensure the availability of NK cells during inflammatory responses. Our findings show that IL-12 is sufficient for the generation of NK-like cells, a process that is also triggered when γc-signalling is functional. To date, IL-15 has been the only cytokine known to act on the progenitor stage to generate mNK cells in mice[12]. We now demonstrate that NKPs also respond to IL-12, leading to the generation of eNK cells. Even though the signalling pathways downstream of IL-12 differ to those triggered by IL-15, both cytokines induce T-bet expression and IFN-γ-production by NK cells[20,35]. However, neither *Tbx21* or *Ifng* are required for the generation of eNK cells (data not shown), and the search for additional factors that control this process is ongoing.

**Figure 2 | IL-12 induces eNK cells in lungs of naive *Rag2*[−/−] *Il2rg*[−/−] mice.** (**a**) Representative plots and absolute numbers of CD45[+]CD3[−] CD122[+]CD49b[+] cells in lungs from IL-12 or Fc-treated (Ctrl) WT or *Rag2*[−/−] *Il2rg*[−/−] mice. The expression of NK1.1 on CD45[+]CD3[−]CD122[+] CD49b[+] cells in lungs from IL-12 or Ctrl-treated WT or *Rag2*[−/−] *Il2rg*[−/−] mice is shown. Data represent five experiments (n ≥ 4 mice per group per experiment). (**b**) Cytospin (×100) of sorted CD45[+]CD3[−]CD122[+]CD49b[+] cells from lungs of IL-12-treated WT or *Rag2*[−/−] *Il2rg*[−/−] mice stained with May–Grünwald–Giemsa solution. Two independent experiments were performed. Scale bar, 20 μm. (**c**) Expression levels of Eomes, NKG2D, NKp46 and DNAM-1 on the NK1.1[+] subsets of CD45[+]CD3[−]CD122[+]CD49b[+] cells isolated from lungs of Ctrl (NK) and IL-12-treated WT mice (NK1.1[+] NK + IL-12), the NK1.1[low] subset of CD45[+]CD3[−]CD122[+]CD49b[+] cells isolated from lungs of IL-12-treated WT mice (NK1.1[low] NK + IL-12) and eNK cells from *Rag2*[−/−] *Il2rg*[−/−] mice. Three independent experiments were performed with at least three mice per group. (**d**) Ki-67 and Bcl-2 expression levels were quantified in lung NK, NK1.1[+] NK + IL-12, NK1.1[low] NK + IL-12 and eNK cells. Data shown represent three independent experiments (n = 3–5 mice per group per experiment). (**e**) Leukocytes isolated from lungs of WT and *Rag2*[−/−] *Il2rg*[−/−] mice either treated with IL-12 or Ctrl were cultured for 4 h with PMA/ionomycin and CD45[+]CD3[−] Gr-1[−] cells were analysed for their IFN-γ production. NK1.1 and NKp46 expression was examined on IFN-γ expressing cells. (**f**) NK cells and eNK cells were isolated from lungs of WT mice and *Rag2*[−/−] *Il2rg*[−/−] mice, respectively. Their cytotoxicity was tested using YAC-1 cells at an effector:target ratio of 6:1. Data are representative of two independent experiments with three mice per group each. Data are expressed as mean ± s.e.m. *P < 0.05, **P < 0.01, ***P < 0.001 as determined by unpaired Student's t-test with Welch's correction or one-way analysis of variance with Bonferroni post-test.

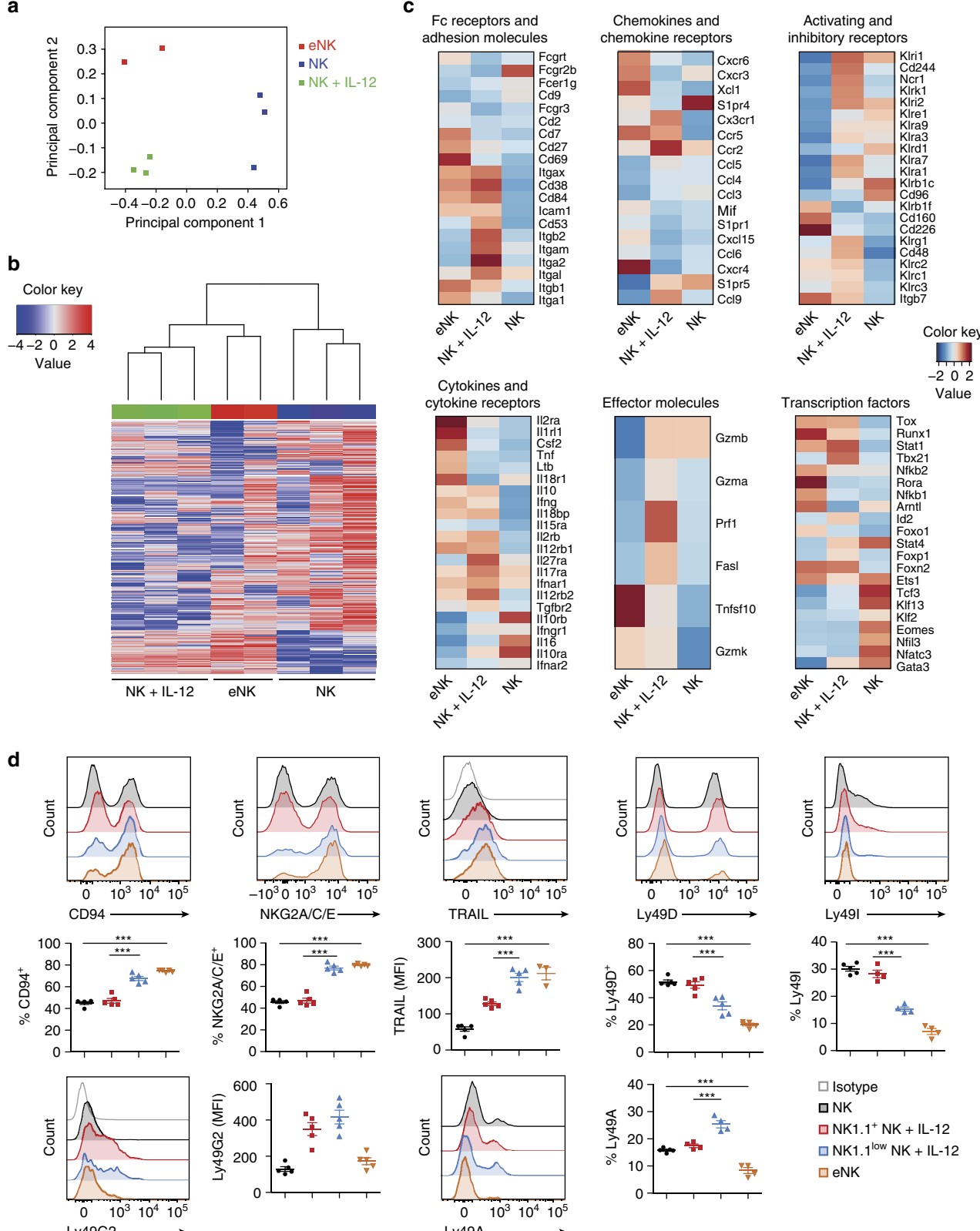

**Figure 3 | Spectrum of distinct and shared genes among different NK-cell subsets. (a)** PCA of gene expression by sorted CD122$^+$CD49b$^+$ cells obtained from WT mice treated with IL-12 (NK + IL-12) or Ctrl (NK) and from IL-12-treated $Rag2^{-/-}Il2rg^{-/-}$ mice (eNK). **(b)** Hierarchical clustering of gene expression by sorted NK + IL-12, eNK and NK cells. **(c)** Heat maps showing differentially expressed genes by sorted eNK, NK + IL-12 and NK cells, clustered to their indicated categories. Heat maps show representative data from one sample from each group. See also Supplementary Table 1 for detailed expression levels and significance. **(d)** Representative histograms and quantification of CD94, NKG2A/C/E, TRAIL, LY49D, Ly49I, Ly49G2 and Ly49A expression on lung NK, NK1.1$^+$ versus NK1.1$^{low}$ NK + IL-12 and eNK cells. Two independent experiments were performed with at least 3–5 mice per group. Data are expressed as mean ± s.e.m. One-way analysis of variance was used to determine significance (***$P < 0.001$). PCA, principal component analyses.

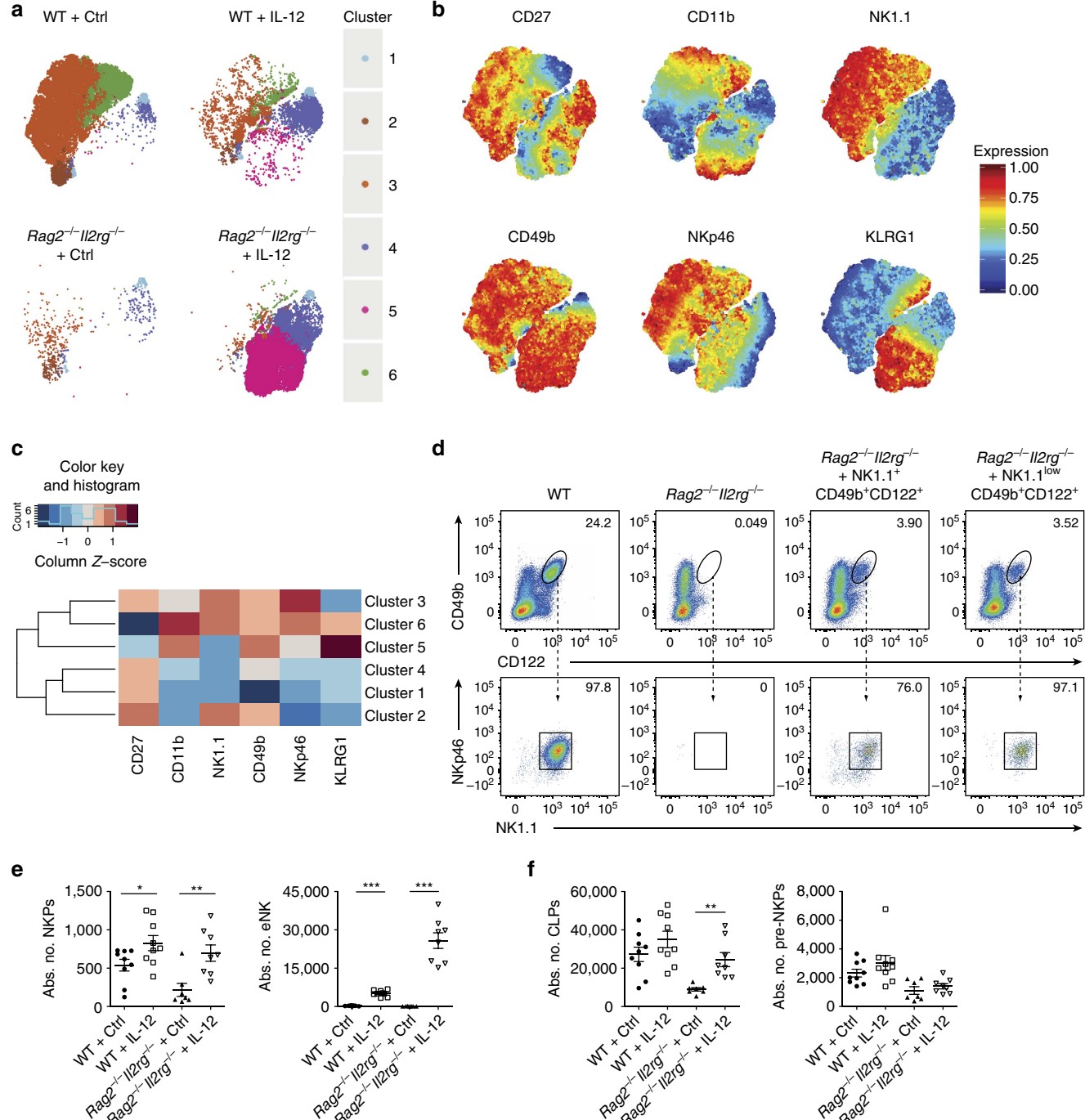

**Figure 4 | IL-12 induces expansion of eNK cells in the BM via an alternative developmental pathway.** (**a**) tSNE dimensions 1 and 2 for CD45$^+$Lin$^-$CD122$^+$ cells derived from the BM of Ctrl- or IL-12-treated WT or $Rag2^{-/-}Il2rg^{-/-}$ mice. (**b**) Annotated t-SNE maps depicting CD27, CD11b, NK1.1, CD49b, NKp46 and KLRG1 in the six identified clusters. (**c**) Heat map summary of average median expression of each cellular marker analysed for six clusters. (**d**) Adoptively transferred splenic CD45$^+$CD3$^-$CD122$^+$CD49b$^+$NK1.1$^+$ and CD45$^+$CD3$^-$CD122$^+$CD49b$^+$NK1.1$^{low}$ cells into $Rag2^{-/-}Il2rg^{-/-}$ mice were analysed in lungs at day 7 after transfer for expression levels of NK1.1. Two independent experiments were performed. (**e**) Quantification of NKPs and eNK cells in the BM of Ctrl- or IL-12-treated WT and $Rag2^{-/-}Il2rg^{-/-}$ mice. Data are pooled from two independent experiments. (**f**) Quantification of CLPs and pre-NKPs in the BM of Ctrl- or IL-12-treated WT and $Rag2^{-/-}Il2rg^{-/-}$ mice. Data are pooled from two independent experiments. Each dot represents one mouse. Data are expressed as mean ± s.e.m. *$P<0.05$, **$P<0.01$, ***$P<0.001$ (unpaired Student's $t$-test with Welch's correction).

NK cells have been classically identified based on their expression of NK1.1 and NKp46 (ref. 22). Thus, the finding that eNK cells only express low levels of both lineage markers initially raised doubts whether eNK cells are actually NK cells. Furthermore, the similarities between NK cells and the group 1 innate lymphoid cells (ILC1)[36,37] could place eNK cells into the latter category. We propose that eNK cells generated by IL-12 described here are *bona fide* NK cells based on: (a) the expression of several lineage defining NK-cell molecules (for example, Eomes), (b) the lack of IL7-R expression and (c) their ability to kill target cells.

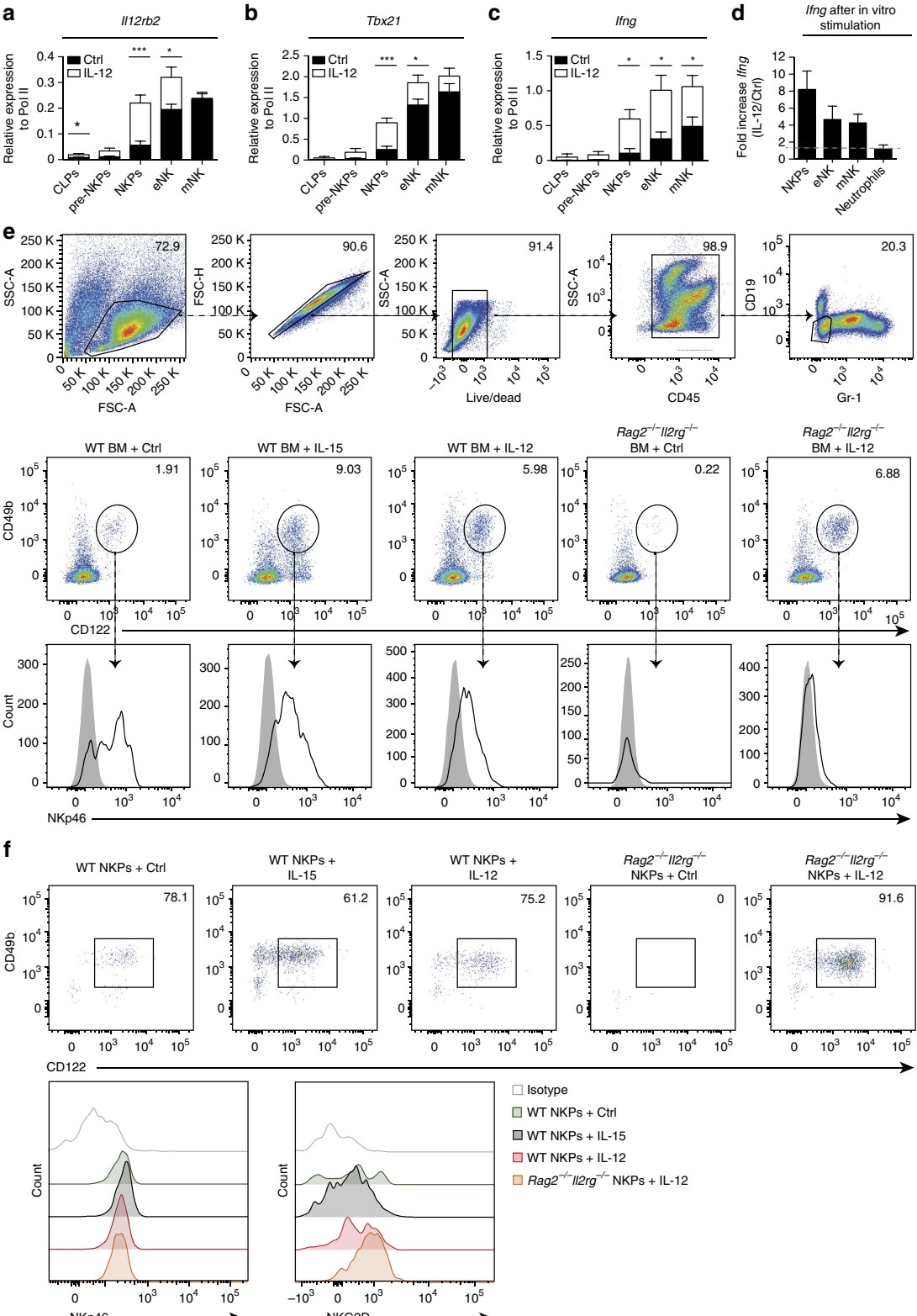

**Figure 5 | *Il12rb2*-expressing NKPs respond to stimulation with IL-12 and differentiate into eNK cells.** CLPs, pre-NKPs, NKPs, eNK and mNK cells were isolated from BM of Ctrl- or IL-12-treated mice and mRNA expression levels of (**a**) *Il12rb2*, (**b**) *Tbx21* and (**c**) *Ifng* were quantified by quantitative reverse transcription PCR (qRT–PCR). Data are shown as pooled samples from three to five mice per group for three independent experiments. (**d**) NKPs, eNK and mNK cells were sorted from WT mice and were treated for 6 h with IL-12 or Ctrl. *Ifng* expression levels were quantified by qRT–PCR. Neutrophils were used as a negative control. Data are shown as pooled samples from three to five mice per group for three independent experiments. (**e**) BM or (**f**) NKPs from WT or *Rag2*$^{-/-}$ *Il2rg*$^{-/-}$ mice were purified and cultured with OP9 stromal cells with Ctrl, with IL-15 or with IL-12 for 4 days before being analysed for differentiation to NK cells. Expression of NKp46 and NKG2D is shown. Data are representative of two independent experiments. Data are expressed as mean ± s.e.m. (*$P < 0.05$, ***$P < 0.001$, unpaired Student's *t*-test with or without Welch's correction).

eNK cells express high amounts of the transcription factor Eomes, required for NK cells to mature past the TRAIL[+] CD11b[low] to the CD49b[+]CD11b[high] stage[38]. Our data indicate that despite being Eomes[+], eNK cells express a wide array of immature markers that resemble TRAIL[+] NK cells found in fetal

and neonatal mice[29]. This NK-cell subpopulation, retained in the liver of adult mice and identified as TRAIL[+] NK1.1[+]CD49b[dim]CD11b[dim]Ly49[−]CD94[+]NKG2[high], exhibits anti-metastatic function against TRAIL-sensitive tumour cells in an IFN-γ-dependent manner[39,40]. Given the high expression of

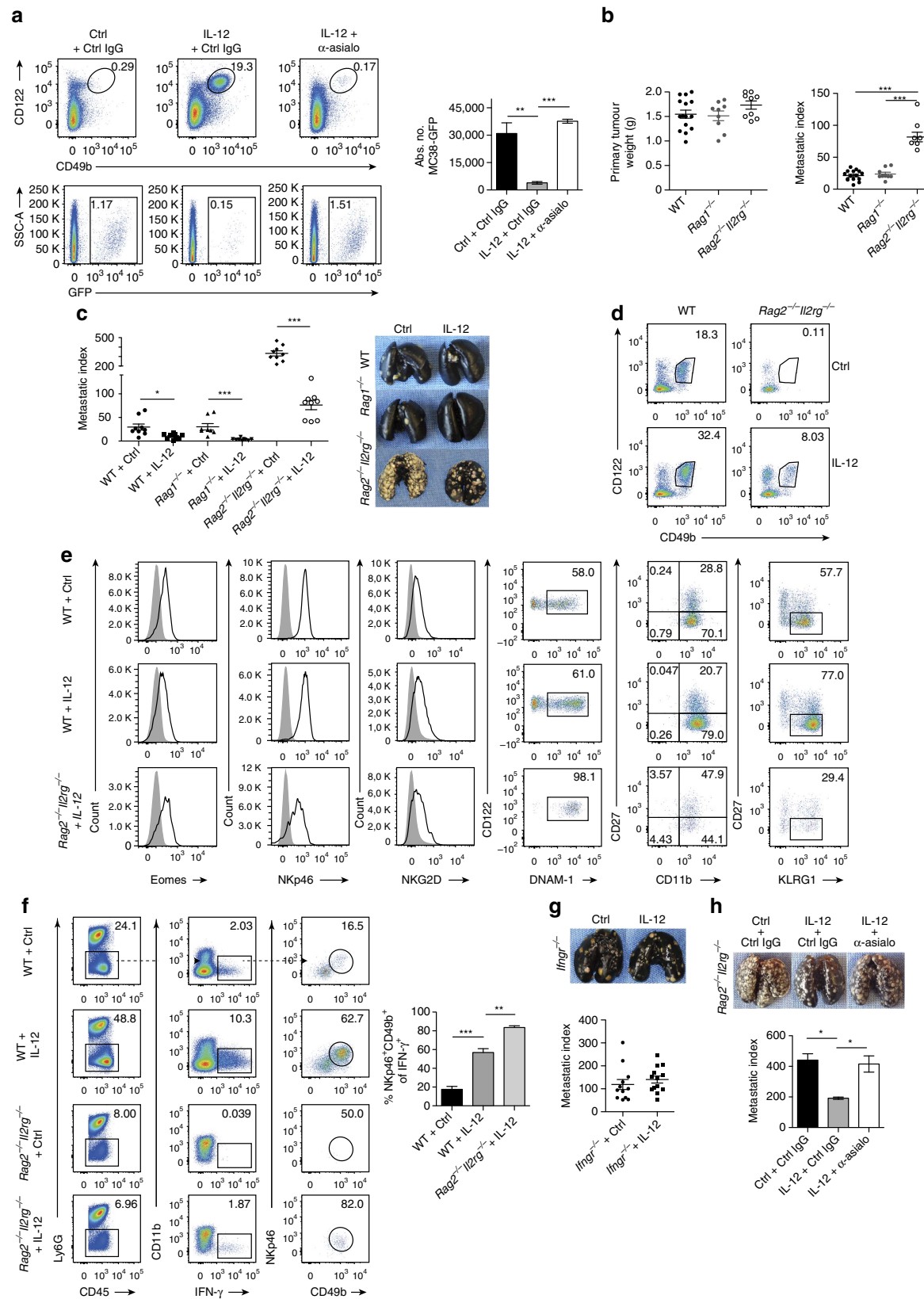

TRAIL in eNK cells and their dependence on IFN-γ to exert metastatic control, it is possible that these two populations utilize similar mechanisms for anti-tumour surveillance. Of note, eNK cells were able to kill two TRAIL-sensitive cell lines such as MC38 and 4T1.

A special feature of eNK cells is their unusually high expression of the activating receptor DNAM-1. Recently, two functionally distinct subsets of splenic NK cells, namely DNAM-1$^+$ and DNAM-1$^-$ NK cells, were described: Highly proliferative DNAM-1$^+$ NK cells show an enforced IL-15 signalling and can give rise to DNAM-1$^-$ NK cells with reduced anti-tumour activity[41]. We propose that the lack of IL-15 signalling arrests eNK cells at a stage characterized by elevated expression of DNAM-1 and a high proliferation rate. Along this line, we attribute the limited expression of NKp46, NK1.1 and Ly49 receptors in these cells to the absence of γc-cytokine signalling. The upregulation of NKp46 and NK1.1 in eNK cells upon adoptive transfer into an IL-15 proficient environment supports this idea. Moreover, the induction of NKp46 and Ly49 on NK cells by IL-15 has been previously reported[42,43]. The high expression of CD49b by NK1.1$^{low}$NKp46$^{low}$ eNK cells is unusual, and suggests that these cells might be generated via an alternative pathway of NK-cell development. NKp46, together with NK1.1, have been shown to be acquired before CD49b in early NK-cell development[25,42]. However, Rosmaraki et al.[44] also identified a population of CD122$^+$NK1.1$^-$CD49b$^+$ cells with lytic activity in the BM of C57BL/6 mice. The mechanisms that regulate CD49b expression during NK-cell development as well as the precise developmental stage in which this marker is acquired will require further investigation.

Collectively, our data present a new paradigm for IL-12-induced emergency NK-cell lymphopoiesis. As such, IL-12 impacts on NKPs to generate a population of unconventional yet functionally competent NK cells. The early production of IL-12 during infections[45,46] suggests that IL-12-induced NK-cell lymphopoiesis occurs in parallel to induction by γc-cytokines to augment host's innate immunity, while shaping the ultimate antigen-specific immune responses. Thus, this pathway may be exploited in patients with severe combined immunodeficiencies (SCID)[47,48]. Since SCID is characterized by markedly reduced numbers of lymphocytes and an increased susceptibility to infection, the generation of eNK cells by a local, controlled delivery of IL-12 might be considered as a potential therapeutic approach for this disease.

## Methods

**Animals.** BALB/c and C57BL/6 mice were obtained from Janvier labs (Roubaix, FR). C57BL/6 $Rag2^{-/-}$ $Il2rg^{-/-}$ and $IL12rb2^{-/-}$ mice were obtained from The Jackson Laboratory. BALB/c $Rag1^{-/-}$ mice were kindly provided by M. Suter (Vetsuisse Faculty, University of Zurich, Switzerland), BALB/c $Rag2^{-/-}$ $Il2rg^{-/-}$ mice by A. Aguzzi (Institute of Neuropathology, University Hospital of Zurich,

Switzerland), BALB/c $Ifngr^{-/-}$ mice by M. Kopf (Institute for Molecular Health Sciences, ETH Zurich, Switzerland) and C57BL/6 $Eomes^{GFP/+}$ mice by Arnold et al.[49]. All animals were kept in house according to institutional guidelines under specific pathogen-free conditions at a 12 h light/dark cycle with food and water provided ad libitum. All experiments were performed using female mice at the age of 7–10 weeks and were performed according to institutional guidelines and approved by the Swiss cantonal veterinary office (license 147/2012 and 142/2015).

**Murine cell lines.** 4T1 cells were kindly provided by M. Detmar (Institute of Pharmaceutical Sciences, ETH, Zurich, Switzerland), YAC-1 cells by M. van den Broek (Institute of Experimental Immunology, University Zurich, Zurich, Switzerland) and MC38-GFP cells by Lubor Borsig (Institute of Physiology, University Zurich, Zurich, Switzerland). OP-9 and 293T cells were purchased from ATCC.

**Expression and purification of IL-12Fc.** As previously described[50], IL-12Fc expressed in 293T cells was purified from supernatant using a protein A column (1 ml, HiTrap, GE Healthcare). After elution with 0.1 M citric acid, pH 3.0 using a purifier (ÄktaPrime) and dialysis for 40 h in PBS, pH 7.4, the concentration and purity of IL-12Fc were measured using the mouse IL-12 (p40) ELISA kit (BD OptEIATM, 555165) and silver staining (Pierce Silver Stain Kit, Thermo Scientific), respectively.

**Orthotopic 4T1 injection.** 4T1 cells ($1 \times 10^5$) were injected in 50 μl PBS in the second mammary fat pad of 7–10 week-old female BALB/c mice. Lung metastases were analysed after ∼23 days. To deplete eNK cells, $Rag2^{-/-}$ $Il2rg^{-/-}$ mice were treated three times with 50 μl of anti-asialo GM1 antibody (Wako Pure Chemical Industries, Japan) starting 7 days after 4T1 tumour inoculation. Lung metastases were quantified at day 18.

**Vaccinia virus infection.** C57BL/6 WT and $Rag2^{-/-}$ $Il2rg^{-/-}$ mice were injected intraperitoneally (i.p.) with $2 \times 10^6$ or $2 \times 10^5$ p.f.u. of the VV strain WR, respectively. $IL12rb2^{-/-}$ mice were injected i.p. with $2 \times 10^6$ p.f.u. Virus was propagated as described[51]. Rat anti-IL-12/23p40 (C17.8, BioXCell) or IgG2A isotype control antibody (2A3, BioXCell) were injected at 500 μg per mouse on day 0 and 2 after VV infection. Mice were taken for analysis 5 days postinfection.

**IL-12 treatment.** At day 7, when a small primary tumour was palpable, 200 ng of IL-12Fc or the IgG fragment as control (Ctrl) diluted in 25 μl PBS were intranasally administered per mouse. Mice were treated three times per week and taken for analysis at day 23 after tumour inoculation. Naive mice were treated three times every 2 days with 200 ng of IL-12Fc or Ctrl IgG diluted in 100 μl PBS intravenously and taken for analysis 6 days after the first injection.

**Quantification of lung metastases.** Pulmonary metastases were quantified by intratracheal injection of India ink (15% India ink in PBS). India ink injected lungs were washed and subsequently placed in Feket's solution (300 ml 70% ethanol, 30 ml 37% formaldehyde and 5 ml glacial acetic acid) overnight. White lung metastases were counted under a dissection microscope. Metastatic index was calculated as the number of lung metastases divided by the primary tumour weight.

**Flow cytometry.** Flow cytometric analysis of lungs from 4T1 tumour-bearing mice was performed at day 23 after tumour cell injection. Lungs were harvested, digested with Collagenase IV (0.4 mg ml$^{-1}$) for 45 min at 37 °C and erythrocytes were lysed with ACK (Ammonium-Chloride-Potassium) lysis buffer. Tibiae and femora were flushed with PBS and erythrocytes were lysed subsequently. Cells were incubated for 20 min in Fc-blocking buffer (2.4G2, 1:1,000, BD Biosciences). The cells were stained with the following antibodies: anti-CD45 (30-

**Figure 6 | Reduced metastases upon induction of eNK cells in $Rag2^{-/-}$ $Il2rg^{-/-}$ mice.** (a) Representative plots and quantification of eNK cells and MC38-GFP cells in lungs of $Rag2^{-/-}$ $Il2rg^{-/-}$ mice treated with control IgG or anti-asialo GM1 antibody (b) Primary tumour weight and metastatic index from WT, $Rag1^{-/-}$ and $Rag2^{-/-}$ $Il2rg^{-/-}$ 4T1 tumour-bearing mice at day 23 after tumour cell injection. Data are representative of two pooled experiments. Each dot represents one mouse. (c) Metastatic index and India-ink-stained lungs from WT, $Rag1^{-/-}$ and $Rag2^{-/-}$ $Il2rg^{-/-}$ 4T1 tumour-bearing mice treated with either IL-12 or Ctrl. Data are pooled from two independent studies. Each dot represents one mouse. (d) CD45$^+$CD3$^-$ leukocytes from lungs of 4T1 tumour-bearing WT or $Rag2^{-/-}$ $Il2rg^{-/-}$ mice treated with IL-12 or Ctrl were analysed for CD122$^+$CD49b$^+$ cells. Data represent three independent experiments ($n \geq 4$ mice per group per experiment). (e) Eomes, NKp46, NKG2D, DNAM-1, CD27, CD11b and KLRG1 expression on CD45$^+$CD3$^-$CD122$^+$CD49b$^+$ NK cells. Data represent three experiments ($n \geq 4$ mice per group per experiment). (f) Representative plot for IFN-γ production of PMA/ionomycin stimulated cells isolated from metastatic lungs of 4T1 tumour-bearing WT and $Rag2^{-/-}$ $Il2rg^{-/-}$ mice treated with either IL-12 or Ctrl. Frequency of NKp46$^+$CD49b$^+$ of total IFN-γ-producing cells. (g) Metastatic index and India-ink-stained lungs from $Ifngr^{-/-}$ 4T1 tumour-bearing mice treated with IL-12 or Ctrl. Pooled data from three experiments with at least four mice per group. (h) Metastatic index and India ink-stained lungs of 4T1 tumour-bearing $Rag2^{-/-}$ $Il2rg^{-/-}$ mice treated with either IL-12 or Ctrl and control IgG or anti-asialo GM1 antibody. Error bars represent ± s.e.m. (a,e) One-way analysis of variance and (b) unpaired Student's t-test with Welch's correction was used to determine significance (*$P < 0.05$; **$P < 0.01$; ***$P < 0.001$).

F11; 1:400, BioLegend), anti-CD49b (Dx5, 1:200, BioLegend), anti-NKp46 (29A1.4, 1:100, eBioscience), anti-CD3 (17A2, 1:100, eBioscience), anti-Gr-1 (6-8C5, 1:400, BioLegend), anti-CD11b (M1/70, 1:200, BioLegend), anti-CD27 (LG.3A10, 1:200, BioLegend), anti-NKG2D (CX5, 1:50, eBioscience), anti-DNAM-1 (10E5, 1:100, BioLegend), anti-CD122 (TM-beta 1, 1:100, BioLegend), anti-CD244.2 (2B4, 1:100, eBioscience), anti-CD127 (A7R34, 1:100, eBioscience), anti-CD135 (A2F10, 1:100, BioLegend), anti-NK1.1 (PK136, 1:200, BD Biosciences), anti-KLRG1 (2F1, 1:100, eBioscience), anti-TRAIL (N2B2, 1:100, BioLegend), anti-Ly49G2 (eBio4D11, 1:100, eBioscience), anti-Ly49A (A1, 1:100, eBioscience), anti-Ly49I (YLI-90, 1:100, eBioscience), anti-NKG2A/C/E (20d5, 1:100, BD Biosciences), anti-CD94 (18d3, 1:100, Biolegend) and anti-Ly49D (4E5, 1:100, Biolegend). To exclude dead cells, we used the Zombie Aqua fixable viability kit (BioLegend). Doublets were excluded by FSC-A/FSC-H gating. For intracellular cytokine staining, cells were stimulated for 4 h at 37 °C and 5% $CO_2$ in RPMI 1640 medium containing 10% FCS, 50 ng ml$^{-1}$ PMA, 500 ng ml$^{-1}$ ionomycin and 1 µl ml$^{-1}$ GolgiPlug (BD Bioscience). For detection of intracellular cytokines, cells were fixed after surface staining and permeabilized with Cytofix/Cytoperm (BD Biosciences) and were then stained with an anti-IFNγ mAb (XMG1.2, 1:100, BD Biosciences). For nuclear stainings, cells were fixed and permeabilized with Fixation/Permeabilization Kit from eBioscience and the following antibodies were used: anti-Eomes (Dan11mag, 1:100, eBioscience), anti-Bcl-2 (BCL/10C4, 1:100, BioLegend) and anti-Ki67 (SOLA15, 1:400, eBioscience). Acquisition was performed on a LSRII Fortessa flow cytometer (BD) and data were analysed using FlowJo Version X (Tree Star).

**Algorithm-guided data analysis.** To equalize the contribution of each marker in subsequent automated data analysis steps, we normalized all data to the 99.9th percentile of the combined dataset, thus preserving intersample variability in maximum expression values, which might be biologically relevant. Following preprocessing, all samples were combined into one file and subjected to the FlowSOM clustering algorithm to identify meaningful immunological populations. After the initial clustering step, resulting nodes were subjected to meta-clustering. The respective k-value was chosen manually to correspond with the according tSNE map. Heatmaps display median expression levels for the indicated populations and were drawn using the ggplot2 R package. Dendrograms were calculated using hierarchical clustering. tSNE maps were calculated using the Rtsne package in R.

**NK-cell and NK-cell precursor isolation.** To isolate CLPs, pre-NKPs, NKPs, mNK and eNK cells, Tibiae and femora were flushed with PBS and erythrocytes were lysed. We enriched for NK cells and their precursors by depletion of CD3$^+$, F4/80$^+$, Gr-1$^+$ (anti-CD3-Bio (145-2C11, 1:100, BioLegend), anti-F4/80-Bio (CI:A3-1, 1:200, Serotec), anti-Gr-1-Bio (RB6-8C4, 1:400, BD Biosciences), Streptavidin Microbeads, Milteny Biotec.) and CD19$^+$ cells (CD19 Microbeads, Milteny Biotec.) using an autoMACS pro-separator (Miltenyi Biotec). CLPs (live, CD45$^+$Lin$^-$CD122$^-$CD127$^+$CD135$^+$CD244$^+$), pre-NKPs (live, CD45$^+$ Lin$^-$CD122$^-$CD127$^+$CD135$^-$CD244$^+$), NKPs (live, CD45$^+$Lin$^-$CD122$^+$ CD244$^+$CD27$^+$CD49b$^-$NKp46$^-$), mNK (live, CD45$^+$Lin$^-$CD122$^+$CD244$^+$ CD49b$^+$NKp46$^+$) and eNK cells (live, CD45$^+$Lin$^-$CD122$^+$CD244$^+$CD49b$^+$ NKp46$^-$) were sorted with an AriaIII Sorter directly into RLT lysis buffer (Qiagen) or into MyeloCult M5300 media if further *in vitro* culturing was required. The following markers were included in the lineage staining (anti-CD3 (17A2, 1:100, BioLegend); anti-CD19 (1D3, 1:200, BD Biosciences); anti-CD14 (mC5-3, 1:400, BD Biosciences); anti-Gr-1 (RB6-8C4, 1:400, eBioscience), anti-CD8 (53-6.7, 1:400, BD Biosciences); anti-CD4 (GK1.5, 1:400, BD Biosciences). The characterization of NKPs was performed according to Fathman *et al.*[30]

***In vitro* culture with IL-12.** Sorted NKPs were cultured in MyeloCult M5300 media (Stem Cell Technologies) with either Fc or IL-12Fc (100 ng ml$^{-1}$) for 6 h before further lysed in RLT buffer (Qiagen). NKPs from BM of WT or $Rag2^{-/-}$ $Il2rg^{-/-}$ mice were cultured on OP-9 cells in 96-well plates in MyeloCult M5300 media (Stem Cell Technologies) with either IL-12 (100 ng ml$^{-1}$), IL-15 (100 ng ml$^{-1}$) or no cytokine. After incubation for 4 days at 37 °C, cells were collected and analysed by flow cytometry. 4T1 cells were seeded in 24-well plates and cultured with either Fc or IL-12Fc (100 ng ml$^{-1}$) for 28 and 48 h. Total number of cells were counted using a Neubauer chamber.

**Adoptive transfer.** CD45$^+$CD3$^-$CD122$^+$CD49b$^+$NK1.1$^{low}$ or NK1.1$^+$ NK cells were isolated from the spleen of IL-12-treated WT mice. 200.000 CD45$^+$ CD3$^-$CD122$^+$CD49b$^+$NK1.1$^{low}$ or NK1.1$^+$ NK cells were injected via the tail vein into $Rag2^{-/-}$ $Il2rg^{-/-}$ mice. The phenotype of the transferred cells was analysed in the lung of these mice 7 days after injection by flow cytometry.

**Quantitative real-time PCR.** RNA was isolated from CLPs, pre-NKPs, NKPs, eNK cells, mNK cells and 4T1 cells using the Qiagen Micro or Mini Kit according to the manufacturer's protocol. Random primers (Invitrogen) were used for synthesis of complementary DNA. The following primers were used for quantitative real-time PCR using a CFX384 Cycler (Bio-Rad Laboratories): Il12rb1

5′-CGCAGCCGAGTAATGTACAAG-3′ and 5′- CGCAGCCGAGTAATGT ACAAG-3′, Il12rb2 5′-TGTGGGGTGGAGATCTCAGT-3′ and 5′-TCTCCT TCCTGGACACATGA-3′, Pol II 5′-CTGGTCCTTCGAATCCGCATC-3′ and 5′-GCTCGATACCCTGCAGGGTCA-3′, Ifng 5′-GCATTC ATGAGTATTGCC AAG-3′ and 5′-GGTGGACCACTCGGATGA-3′, Tbet 5′-CAACAACCCCTT TGCCAAAG-3′ and 5′-TCCCCCAAGCAGTTGACAGT-3′. Subsequent analyses were performed with Excel calculating the dCt values.

**Next-generation sequencing.** Complementary DNA libraries were generated from RNA purified from NK-cell populations, amplified using the SMART-seq2 Amplification Kit (Clontech), and sequenced for 200–250 million reads using 50 bp paired-end at the Quantitative Genomics Facility in Basel. Reads were quality-checked with FastQC. Low-quality ends were clipped (3 bases from the start, 10 bases from the end). Trimmed reads were aligned to the reference genome and transcriptome (FASTA and GTF files, respectively, downloaded from the UCSC mm10 repository) with STAR version 2.3.0e_r291 (ref. 52) with default settings. Distribution of the reads across genomic isoform expression was quantified using the R package GenomicRanges[53] from Bioconductor Version 3.0. Differentially expressed genes were identified using the R package edgeR[54] from Bioconductor Version 3.0. Sequencing information is available at the European Bioinformatics Institute (EBI; ENA: PRJEB15668).

**Killing assays.** Lung NK cells (live, CD45$^+$CD3$^-$CD122$^+$CD49b$^+$) were sorted with an BD FACSAria III sorter. YAC-1 cells were stained with PKH26 Red Fluorescent Cell Linker Mini Kit (Sigma, MINI26-1KT). NK cells were incubated with YAC-1 target cells at an effector:target ratio of 6:1 in RPMI supplemented with 10% FCS for 5 h at 37 °C in 5% $CO_2$. After removal of medium, Topro (0.8 µM) was added to the cells and cells were acquired on a LSRII Fortessa flow cytometer (BD). Data were analysed using FlowJo Version X (Tree Star). The percentage specific lysis was calculated as followed: [(Experimental lysis— spontaneous lysis)/(maximum lysis—spontaneous lysis)] × 100%.

For *in vivo* killing assays, $Rag2^{-/-}$ $Il2rg^{-/-}$ mice were injected three times every two days with 200 ng of IL-12Fc. eNK cell depletion was performed by injecting 50 µl of anti-asialo GM1 antiserum (Wako Pure Chemical Industries, Japan) twice starting 1 day before IL-12Fc treatment. Together with the third injection of IL-12Fc, mice were injected with $2 \times 10^6$ MC38-GFP tumour cells. Twenty-four hours later, animals were perfused with PBS and lungs were removed. The percentages and total numbers of MC38-GFP cells and CD49b$^+$CD122$^+$ cells in the lungs was analysed by flow cytometry.

**Cytospin and May–Grünwald–Giemsa staining.** Morphological analysis was performed by cytospin ($1 \times 10^5$ cells per sample) followed by May–Grünwald–Giemsa staining. Images were taken with an Olympus BX41 microscope using the cell^B software (Version 2.8).

**Enzyme-linked immunosorbant analysis.** IL-12 levels were measured using a mouse IL-12 (p40) ELISA kit according to the user manual (BD OptEIA™, 555165).

**Statistics.** $P$ values were calculated using GraphPad statistical software (GraphPad Software Inc.). $P$ values $< 0.05$ were considered significant. *$P < 0.05$; **$P < 0.01$; ***$P < 0.001$. If no stars are indicated, no statistically significant difference was found.

**Data availability.** Sequencing information is available at the European Bioinformatics Institute (EBI; ENA: PRJEB15668). The authors declare that all data are available within the Article and its Supplementary Information files, or are available from the author upon request.

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

## Acknowledgements

We thank Sabrina Nemetz and the Flow Cytometry Facility, University of Zurich for technical assistance. This work was supported by grants from the Swiss National Science Foundation (310030_146130 and 316030_150768 to B.B.), the University Research Priority Project 'Translational Cancer Research' (B.B., S.T. and M.v.d.B), and the Forschungskredit of the University of Zurich (grant no. FK-14-024 to I.O.).

## Author contributions

I.O., S.T., M.v.d.B. and B.B. designed research, I.O., S.T., K.N. and M.v.d.B. performed the research and analysed the data. C.M. helped to design studies and interpreted the data, S.A. provided mice, S.A.Q. participated in discussing the paper and I.O., S.T. and B.B. wrote the manuscript.

## Additional information

**Competing financial interests:** The authors declare no competing financial interests.

DOI: 10.1038/ncomms15185    OPEN

# Corrigendum: Interleukin-12 bypasses common gamma-chain signalling in emergency natural killer cell lymphopoiesis

Isabel Ohs, Maries van den Broek, Kathrin Nussbaum, Christian Münz, Sebastian J. Arnold, Sergio A. Quezada, Sonia Tugues & Burkhard Becher

*Nature Communications* 7:13708 doi: 10.1038/ncomms13708 (2016); Published 16 Dec 2016; Updated 4 Apr 2017

The financial support for this Article was not fully acknowledged. The acknowledgements should have included the following:

The European Community FP7 grant no. 602239 (ATECT).

