## [Peer Review File · Nature Communications]

Reviewers' comments:

Reviewer #1, expert in NK development (Remarks to the Author):

This is a high quality, very interesting and technically proficient paper from the Becher lab that postulates a novel pathway for IL-12 driven NK cell development.

Comments:

1. The authors state (p.3) that the NKPs are "normal" in various mutant mice strains when the proper characterization of NKPs was only published by Carotta et al., and Fathman et al., in 2010/11, are all the 3 referenced papers really referring to NKPs?
2. I am surprised not to see any mention in the paper of the data presented by the Sun Lab in 2013 (Firth et al, JEM) that shows IL-12 driven NK cell expansion post MCMV infection.
3. I don't entirely understand how in Figure 1A, the authors are showing a 25-fold change for CD122+CD49b+ cells following VV infection in the dot plot data but this seems to be barely a 2-fold change in absolute numbers.
4. Again in Figure 5F, I fully appreciate the variability in cell numbers and the technical difficulty but the dot plots don't really reflect the percentages given.
5. As the authors realise, it's not straightforward to call something a NK cell that expresses such low levels of NKp46 and NK1.1, the expression of KIRs is a fairly fundamental property of NK cells, have the authors looked at them relative to cNKs.
6. There is neither a clear explanation for its use nor any references given for the 4T1 model of breast cancer (p.10). There is a statement (p.10) that "control of metastasis depends entirely on NK cells since Rag2-/-Il2rg-/- mice showed an increased metastatic burden in comparison to WT and Rag1-/- mice", is it therefore possible that your observed effect on metastasis has nothing to do with eNK cells at all?
7. IL-12 KO and IL-12R KO mice (e.g. those generated by Jeanne Mangan) are reported to have "normal" NK cell numbers. Perhaps the authors should confirm this for themselves and, if correct, could they account for these data given their proposed role for IL-12 on NKPs.

Reviewer #2, expert in mass cytometry (Remarks to the Author):

Ohs et al report a novel class of Natural Killer (NK) cells which mature based on exposure to IL-12 rather than the conventional γ c-dependent cytokines, e.g. IL-15. Their observations are based on responses to vaccinia virus infection of murine Rag2-/-IL2rg-/- mice in which NK cell lymphopoiesis was detected. They authors found that IL-12-driven NK cells expressed markers of NK cells (eomes and NKG2D) although altered levels of other NK cell markers (low NKp46, NK1.1, and high DNAM-1). The γ c-independent unconventional NK cells, named eNK, produced IFN γ and were cytotoxic. Transcriptional analysis showed the unconventional NK cells resembled NK cells treated with IL-12 although they were less mature and showed distinct patterns of cytokine production and effector mechanisms. eNK cells were also active in tumor rejection in a murine model of metastatic 4T1 breast cancer. The authors conclude that IL-12/IL-15 action on NK cell precursors in the bone marrow lead to different maturation of the cell types with NK-like markers and distinct functional effector mechanisms. The manuscript is well written and clearly presented and the conclusions are supported by the data provided.

1. The authors demonstrate the presence of a distinct class of NK cells that develop in response to IL-12. While they are distinct, and perhaps unconventional, it is not clear why they would be considered 'emergency'. This pathway exists in WT mice as well and may be relevant in particular infectious settings.
2. Does different cytokine production and killing efficiency indicate a niche for immune responses (lower granzymes and higher cytokines) rather than a step in differentiation program? These unconventional NK cells bypass canonical γ c-chain signaling and possibly have specialized anti-tumor function.

3. The proposed use of IL-12 in human therapeutics is quite speculative.

Reviewer #3, expert in anti-tumor NK responses (Remarks to the Author):

Ohs et al. report the existence of lymphocytes termed eNK cells that resemble immature NK cells in IL-12-injected Rag^{-/-}Il2rg^{-/-} mice. They are CD122⁺CD49b⁺NK1.1^{-/lo}, highly express CD69, Cxcr6, Tnf, ST2, CD25, Csf2 (GM-CSF), Ltb and Tnfsf10 (TRAIL) and produce IFN γ . They may develop from NKPs upon IL-12 stimulation. They also appeared to have anti-metastasis effects in vivo.

General comment

As the authors state, the presence of NK cells in Rag^{-/-}Il2rg^{-/-} mice and possible role of IL-12 in their development or expansion have previously been reported by Sun et al. Similarly, NK cells have been detected in IL-15Ra-deficient mice (J. Exp Med. 197:977, 2003). The finding that the Il2rg-independent eNK cells develop from NKPs by IL-12 is novel. The gene expression profiles of eNK cells and possible anti-metastatic functions of eNK cells are also novel. On the other hand, the functional importance of eNK cells in normal mice is unclear. Although immature NK cells resembling eNK cells were detected in IL-12-injected WT mice, they were not characterized in sufficient detail to make direct comparison with eNK cells.

The authors should provide some key information that is missing in this paper, as discussed below.

Specific comments

1. Experimental procedures are not described in sufficient details. For example, how much IL-12 was injected, how many times in what route, and how long after IL-12 injection were the mice analyzed. The age of the mice used in this study should also be stated.
2. In Figure 1, how many days after the infection of VV were NK cells analyzed? Do WT and Rag^{-/-}Il2rg^{-/-} mice differ in the kinetics of IL-12 production and eNK cell generation?
3. In the flow analyses in Figure 2, the authors should gate for the NK1.1⁻ subset of CD49b⁺CD122⁺ cells from WT and Rag^{-/-}Il2rg^{-/-} mice and analyze more extensive marker expression including CD94 and Ly49 as they are important markers to differentiate immature and mature NK cells.
4. In the transcriptome analyses (Figure 3), NK1.1^{low} cells from IL-12-injected WT mice should be purified and compared with those from RAG^{-/-} γ c^{-/-} mice. Because eNK-like cells in WT mice are rather rare, it is difficult to directly compare the same populations in WT and RAG^{-/-} γ c^{-/-} mice.
5. The authors state that eNK cells express Klr (p7). However, Figure 3 shows very low, if any, expression of most Klr genes. As discussed above, the expression of KLRs should be tested by flow cytometry.
6. The differentiation of NKPs into eNK cells in vitro in Figure 5 is not very convincing. Figure 5f only shows upregulation of CD49b on NKPs (CD122⁺CD49b⁻) by IL-12 stimulation in vitro. Whether they acquire NKp46. and NKG2D should also be tested.
7. Anti-metastasis effects of IL-12 injection have been reported previously. Although the results presented in Figure 6 support the authors' contention that IL-12-induced eNK cells may play a major role, authors should test the effects of eNK cell depletion for more conclusive results. The data in Figure 6f are difficult to interpret without proper controls. Ideally Ifng^{-/-} Rag^{-/-}Il2rg^{-/-} mice should be tested.
8. Supplementary table 1 should include transcription factors known to be important for NK cell development, e.g. Nfil3 and Id2.

Reviewer #4, expert in NK biology (Remarks to the Author):

Oh's and colleagues identify an alternative pathway of NK cell development driven by the proinflammatory cytokine IL-12, which can occur independently of common gamma chain signaling. Administration of IL-12 or induction of this cytokine by virus infection was sufficient to elicit the emergence of a population of unique NK cells by specifically targeting NK cell precursors (NKPs) in the bone marrow (BM). These unique NK cells were able to limit metastatic growth in a mouse model of breast cancer. These data reveal a novel IL-12-driven hard-wired pathway of emergency NK lymphopoiesis bypassing steady state common gamma chain signaling from cytokines like IL-15. The findings are novel and advance the field by revealing potential therapeutic means of inducing NK cells in SCID, NK cell-deficient, or other patients.

Major issues:

Too little data is provided to effectively distinguish these eNK from ILC1. Namely, the staining for CD127 expression that is described should be shown in the manuscript, and other defined markers that distinguish these cells should be explored if possible. The YAC-1 cell killing in Figure 2F is very unconvincing. The data shown is unclear with regards to how clean the staining with Topro was and whether YAC-1 cells were lost to cell death, the protocol for Topro staining and interpretation of results is poorly described, and the assay is performed only at low E:T ratios. There are more accurate means of measuring NK cell killing of YAC-1 cells than that used by the authors (i.e. chromium release).

How can the authors be certain that IL-12 in Figure 6 is not directly acting on the tumor? What evidence is there that the eNK cells are required for this anti-metastatic effect?

This reviewer was not clear from the experiments shown or the authors' interpretation whether these eNK cells were entirely *de novo* generated following infection or administration of IL-12, or whether a small subset of these cells was simply expanded. Clearly the effect of IL-12 *in vitro* can drive differentiation of NKPs, but is this the major mechanism *in vivo*. Maybe data from uninfected mice in Figure 1 demonstrating absence of these cells would be helpful.

At the top of page 5 and in other places, the conclusion that IL-12 is the main or only cytokine ("our findings exclude the requirement of other factors besides IL-12...", page 11) driving this emergency lymphopoiesis is not rigorously tested since no other cytokines are examined, co-contributions from supporting cytokines are not addressed, and since blockade experiments do not distinguish between IL-12 (p40 homodimers or p70) and IL-23.

Minor issues:

Experimental details are missing: 1) duration of VV infection prior to analysis, 2) which species of IL-12 is measured in Figure 1B, 3) what species of IL-12 is administered *in vivo*, and 4) how many experiments constitute 'multiple' in the Figure 2A legend.

Why was the presence of NK cells in the liver or other tissues not examined?

In addition to the representative data shown in Figure 2C, it would be useful to provide graphs revealing the mean and deviation of these various parameters (proportion and MFI of DNAM-1, etc.) across multiple mice.

The authors should stick with one naming convention and avoid the switch from eNK to NK-IL-12 in Figure 5.

This reviewer finds no demonstration of unbridled protective properties in the data that would substantiate the conclusion drawn by the authors on page 6.

Point-by-point response

Reviewer #1, expert in NK development (Remarks to the Author):

This is a high quality, very interesting and technically proficient paper from the Becher lab that postulates a novel pathway for IL-12 driven NK cell development.

Comments:

1. The authors state (p.3) that the NKPs are "normal" in various mutant mice strains when the proper characterization of NKPs was only published by Carotta et al., and Fathman et al., in 2010/11, are all the 3 referenced papers really referring to NKPs?

The characterization of NKPs performed by Vosshenrich et al. in 2005 described NKPs as bone marrow $CD122^+CD49b^-NK1.1^-$ cells. The later reports by Carotta et al. 2011 and Fathman et al. 2011, however, included additional cell surface markers to better characterize NKPs, such as CD244 and CD27. We used these markers to unambiguously identify NKPs and found reduced numbers of $Lin^+CD122^+CD27^+CD244^+NK1.1^-CD49b^-$ NKPs in $Rag2^{-/-}Il2rg^{-/-}$ mice in comparison to WT mice (See Fig. 4e and attached graph from another independent experiment). Fathman et al., 2011 has been added to the Material and Methods section.

Quantification of NKPs in the BM of WT and $Rag2^{-/-}Il2rg^{-/-}$ mice. NKPs were gated as $Lin^+CD122^+CD27^+CD244^+NK1.1^-CD49b^-$ cells. Each dot represents one mouse. $*p < 0.05$, (unpaired Student's t-test with Welch's correction).

2. I am surprised not to see any mention in the paper of the data presented by the Sun Lab in 2013 (Firth et al, JEM) that shows IL-12 driven NK cell expansion post MCMV infection.

Indeed, the paper published by the Sun lab in 2013 was referred to several times in our manuscript but was not cited in the context of IL-12 driven NK cell expansion upon MCMV infection. We have expanded our introduction (page 3) accordingly.

3. I don't entirely understand how in Figure 1A, the authors are showing a 25-fold change for CD122⁺CD49b⁺ cells following VV infection in the dot plot data but this seems to be barely a 2-fold change in absolute numbers.

We are glad the reviewer picked up this error. In order to discover the reason for the differences between fold change and absolute numbers of CD122⁺CD49b⁺ cells, we carefully reanalyzed our FACS data files in Fig. 1a. We found that the CD122⁺CD49b⁺ gate in lungs of PBS-treated *Rag2^{-/-}Il2rg^{-/-}* mice (0.70% of CD45⁺CD3⁻ cells) contained a small proportion of contaminating cells, possibly of myeloid origin expressing CD49b (described by Hamaguchi-Tsuru, Br J Haematol 2004). After gating out these cells, we found virtually no CD49b⁺CD122⁺ cells in lungs of PBS-treated *Rag2^{-/-}Il2rg^{-/-}* mice (See new Fig. 1a). This resulted in an increase of about 100-fold absolute numbers of CD49b⁺CD122⁺ cells in VV-infected compared to PBS-treated *Rag2^{-/-}Il2rg^{-/-}* mice. We have attached a table (see below) displaying total numbers of CD45⁺ leukocytes and CD122⁺CD49b⁺ cells in PBS-treated and VV-infected *Rag2^{-/-}Il2rg^{-/-}* mice, as well as the corresponding amounts of counting beads present in each sample for the calculation of absolute cell numbers.

Counting beads contained 1090 beads/ul and were used 1/10

Sample:	Used for quality control of beads			CD45+SSC low	CD122+CD49b+ Counts	% CD122+CD49b+	Total CD122+CD49b+stained	Lung was divided by 3
	Beads count	Bead ratios	Bead ratios					
WT PBS 1	1159	56.3	43.7	174820	17266	10.1412	162380.8456	487142.5367
WT PBS 2	1304	51.8	48	202369	19279	9.6084	161151.1503	483453.4509
WT PBS 3	841	50.2	49.8	146579	16212	11.2832	210119.8573	630359.5719
WT VV 1	430	51.2	48.8	96521	4483	4.8124	113638.8372	340916.5116
WT VV 2	643	47.7	52.3	191879	20259	10.8794	343426.285	1030278.849
WT VV 3	759	53	47	169079	18378	11.1573	263926.4822	791779.4466
Rag2^{-/-}Il2rg^{-/-} PBS 1	811	50.8	49.2	36750	6	0.017507	80.64118372	241.9235512
Rag2^{-/-}Il2rg^{-/-} PBS 2	623	51.7	48.3	29867	5	0.017287	87.47993579	262.4398074
Rag2^{-/-}Il2rg^{-/-} PBS 3	686	50.7	49.3	34562	10	0.03102	158.8921283	476.6763848
Rag2^{-/-}Il2rg^{-/-} VV 1	1068	49.2	50.7	16315	679	4.4156	6929.868914	20789.60674
Rag2^{-/-}Il2rg^{-/-} VV 2	984	50.7	49.3	15329	806	5.512	8928.252033	26784.7561
Rag2^{-/-}Il2rg^{-/-} VV 3	1041	49.7	50.2	17903	687	4.14505	7193.371758	21580.11527

4. Again in Figure 5F, I fully appreciate the variability in cell numbers and the technical difficulty but the dot plots don't really reflect the percentages given.

We have carefully revised Fig. 5f and the plots reflect the percentages given. The differentiation of NKPs from *Rag2^{-/-}Il2rg^{-/-}* mice, for example, did not give rise to CD122⁺CD49b⁺ cells (0% indicated in the plots), but both treatments with IL-12 and IL-15 lead to the generation of high percentages of CD122⁺CD49b⁺ cells (between 60 and 92% depending on the treatment).

5. As the authors realise, it's not straightforward to call something a NK cell that expresses such low levels of NKp46 and NK1.1, the expression of KIRs is a fairly fundamental property of NK cells, have the authors looked at them relative to cNKs.

We agree. Therefore, we have analyzed the expression of several Ly49 receptors (homologues of KIRs in mice) on eNK cells from *Rag2^{-/-}Il2rg^{-/-}* mice. Of note, several members of the Ly49 receptor family such as Ly49A (*Klra1*), Ly49D (*Klra4*) and Ly49I (*Klra9*) were found to be expressed in eNK cells (albeit at lower levels than in CD49b⁺CD122⁺ cells emerging in Ctrl- or IL-12-treated WT mice) (See **new Fig. 3d**). Moreover, the expression of Ly49D and Ly49I in NK^{low} CD49b⁺CD122⁺ cells in IL-12-treated WT mice (NK^{low} NK + IL-12) was similar to that found on eNK cells from *Rag2^{-/-}Il2rg^{-/-}* mice, thus solidifying the concept that IL-12-mediated emergency lymphopoiesis also occurs in the face of physiological γ c-signaling.

6. There is neither a clear explanation for its use nor any references given for the 4T1 model of breast cancer (p.10). There is a statement (p.10) that "control of metastasis depends entirely on NK cells since *Rag2^{-/-}Il2rg^{-/-}* mice showed an increased metastatic burden in comparison to WT and *Rag1^{-/-}* mice", is it therefore possible that your observed effect on metastasis has nothing to do with eNK cells at all?

We apologize for not having clarified this point better. The 4T1 model is a clinically relevant model of breast cancer that frequently metastasizes to the lung (Aslakson *et al.*, Cancer Research 1992). We used this model to determine the anti-tumor properties of eNK cells, and more specifically to investigate whether they could control metastatic spread. The key role of ILCs in the control of lung metastasis is given by the fact that *Rag2^{-/-}Il2rg^{-/-}* mice (lacking all lymphocytes) and devoid of eNK cells showed a drastic increase in the number of metastasis compared to *Rag^{-/-}* (lacking B and T lymphocytes) and WT mice (Fig. 6a). Unexpectedly, IL-12 was still able to reduce metastasis in *Rag2^{-/-}Il2rg^{-/-}* mice, showing that yet another cell type could respond to IL-12. We analyzed the production of Ifn γ , the main target of IL-12, to track the responder cell type to IL-12 in *Rag2^{-/-}Il2rg^{-/-}* mice, which is how we initially discovered eNK cells (Fig. 6f). The dependence of IL-12-induced metastatic suppression on eNK cells was demonstrated through their depletion using NK cell depletion with an anti-asialo-GM1 antibody (See **new Fig. 6h** and revised **pages 10 and 11** of the Results section).

7. IL-12 KO and IL-12R KO mice (e.g. those generated by Jeanne Mangan) are reported to have "normal" NK cell numbers. Perhaps the authors should confirm this for themselves and, if correct, could they account for these data given their proposed role for IL-12 on NKPs.

This is an interesting point. To address this, we quantified total NK cell numbers in WT and *IL-12R β 2^{-/-}* mice and found normal numbers of CD122⁺CD49b⁺NK1.1⁺ NK cells in lungs of *IL-12R β 2^{-/-}* mice in comparison to WT mice, in agreement with previous data published on *IL-12R β 2^{-/-}* and *IL-12R β 2^{-/-}* mice (Magrath *et al.*, Immunity 1996, Wu *et al.*, J. Immunol 2000). This makes sense considering the absence/low levels of circulating IL-12 found in steady state. This also strengthens the notion that IL-12 mediates emergency NK lymphopoiesis, but not steady state homeostasis. A

file showing the quantification of total lung NK cells in WT and *IL-12Rβ2*^{-/-} mice has been added below.

Quantification of NK cells in the BM of WT and *IL-12Rβ2*^{-/-} mice. NK cells were gated as CD122⁺NK1.1⁺NKp46⁺CD49b⁺ cells. Each dot represents one mouse.

Reviewer #2, expert in mass cytometry (Remarks to the Author):

Ohs et al report a novel class of Natural Killer (NK) cells which mature based on exposure to IL-12 rather than the conventional γ c-dependent cytokines, e.g. IL-15. Their observations are based on responses to vaccinia virus infection of murine Rag2^{-/-}IL2rg^{-/-} mice in which NK cell lymphopoiesis was detected. They authors found that IL-12-driven NK cells expressed markers of NK cells (eomes and NKG2D) although altered levels of other NK cell markers (low NKp46, NK1.1, and high DNAM-1). The γ c-independent unconventional NK cells, named eNK, produced IFN γ and were cytotoxic. Transcriptional analysis showed the unconventional NK cells resembled NK cells treated with IL-12 although they were less mature and showed distinct patterns of cytokine production and effector mechanisms. eNK cells were also active in tumor rejection in a murine model of metastatic 4T1 breast cancer. The authors conclude that IL-12/IL-15 action on NK cell precursors in the bone marrow lead to different maturation of the cell types with NK-like markers and distinct functional effector mechanisms. The manuscript is well written and clearly presented and the conclusions are supported by the data provided.

We thank the reviewer for the astute summary and positive evaluation.

1. The authors demonstrate the presence of a distinct class of NK cells that develop in response to IL-12. While they are distinct, and perhaps unconventional, it is not clear why they would be

considered 'emergency'. This pathway exists in WT mice as well and may be relevant in particular infectious settings.

We appreciate the comment and have therefore better clarified the term “emergency NK cell lymphopoiesis”. We expect eNK cells to be only generated under inflammatory conditions and in response to IL-12. This might be the case for several viral and bacterial infections (Byron *et al.*, Current Opinion Immunology 1997, Nguyen *et al.*, J Immunol 2002), in which IL-12 production by activated myeloid cells would lead to eNK lymphopoiesis in parallel to conventional NK cell induction by γ c-cytokines.

As for antitumor responses, eNK cells could be elicited not only upon direct IL-12 treatment, but also in response to any anti-cancer therapy leading to a strong activation of myeloid cells, such as vaccines or CD40L (Carrant *et al.*, Molecular Therapy 2015). We hypothesize that this emergency NK lymphopoiesis induced by IL-12 might be required in both pathological contexts to satisfy the high demand of NK cells required for an efficient innate immune response. We have clarified this point in the revised version of the manuscript (see **discussion**).

2. Does different cytokine production and killing efficiency indicate a niche for immune responses (lower granzymes and higher cytokines) rather than a step in differentiation program? These unconventional NK cells bypass canonical γ c-chain signaling and possibly have specialized anti-tumor function. Can we answer to that?

This is an interesting point, which we investigated in some detail and included in the revised version. We think that eNK cells develop via an alternative differentiation pathway in which they upregulate CD49b before expressing the lineage markers NK1.1 and NKp46 (Fig. 4a-c). This is in contrast to the conventional pathway of NK cell development, in which NK cells develop from CD49b⁻NK1.1⁻NKp46⁺ to CD49b⁻NK1.1⁺NKp46⁺ and finally to CD49b⁺NK1.1⁺NKp46⁺ cells (Liu *et al.*, Trends Immunology 2013). We have conducted additional experiments to investigate the anti-tumor function of eNK cells (See **new Fig. 6a**). Due to the high levels of TRAIL observed in eNK cells (Fig. 3), the capacity of these cells to eliminate tumor cells was tested using the TRAIL-sensitive MC38 colon carcinoma cell line. Intravenously inoculated MC38-GFP cells were efficiently eliminated from lungs of IL-12-treated *Rag2^{-/-}Il2rg^{-/-}* mice, a process which was shown to be completely dependent on eNK cells (See **new Fig. 6a**). These experiments, together with the role of eNK cells in metastasis control (Fig. 6) highlight the key role of this population in tumor immune surveillance.

3. The proposed use of IL-12 in human therapeutics is quite speculative. We propose that a local targeted use of IL-12 could be beneficial to induce eNK cells in lymphopenic patients. Alternatively, cancer patients might also profit from the induction of eNK cells by IL-12, due to the above-mentioned antitumor activity of this cell population. Even though potent antitumor properties of IL-12 were observed more than 20 years ago in various transplantable cancer models, the majority of clinical trials involving IL-12 treatment were associated to toxic side effects (Tugues

et al., Cell Death and Differentiation 2015). Therefore, the generation of eNK cells in this pathological context would only be beneficial using a targeted and controlled strategy to locally deliver IL-12. If so requested, we can of course also remove this part of the discussion, but we felt that this should be included. We adjusted the language to better indicate that we are speculating.

Reviewer #3, expert in anti-tumor NK responses (Remarks to the Author):

Ohs et al. report the existence of lymphocytes termed eNK cells that resemble immature NK cells in IL-12-injected Rag^{-/-}Il2rg^{-/-} mice. They are CD122⁺CD49b⁺NK1.1^{-/lo}, highly express CD69, Cxcr6, Tnf, ST2, CD25, Csf2 (GM-CSF), Ltb and Tnfsf10 (TRAIL) and produce IFN γ . They may develop from NKPs upon IL-12 stimulation. They also appeared to have anti-metastasis effects in vivo.

General comment

As the authors state, the presence of NK cells in Rag^{-/-}Il2rg^{-/-} mice and possible role of IL-12 in their development or expansion have previously been reported by Sun et al. Similarly, NK cells have been detected in IL-15Ra-deficient mice (J. Exp Med. 197:977, 2003). The finding that the Il2rg-independent eNK cells develop from NKPs by IL-12 is novel. The gene expression profiles of eNK cells and possible anti-metastatic functions of eNK cells are also novel. On the other hand, the functional importance of eNK cells in normal mice is unclear. Although immature NK cells resembling eNK cells were detected in IL-12-injected WT mice, they were not characterized in sufficient detail to make direct comparison with eNK cells.

The authors should provide some key information that is missing in this paper, as discussed below.

Specific comments

1. Experimental procedures are not described in sufficient details. For example, how much IL-12 was injected, how many times in what route, and how long after IL-12 injection were the mice analyzed. The age of the mice used in this study should also be stated.

We have carefully revised the manuscript and added all the missing information on how much IL-12 was injected, the frequency and the route of administration, as well as the time when the mice were analyzed (See **Material and Methods-IL-12 treatment**). The age of the mice used in the study has also been added in the Material and Methods part (See **Material and Methods-Animals**).

2. In Figure 1, how many days after the infection of VV were NK cells analyzed? Do WT and Rag^{-/-}Il2rg^{-/-} mice differ in the kinetics of IL-12 production and eNK cell generation?

Mice were taken for analysis 5 days after VV infection. This information has now been incorporated in Figure 1 (see **new Fig. 1c**) and in the Material and Methods section (see **Material and Methods-Vaccinia virus infection**). At this time point, IL-12 production in the serum was analyzed (Fig. 1b) and was detected in high levels in both VV-infected WT and Rag2^{-/-}Il2rg^{-/-} mice. Serum levels of IL-12

were, however, lower in VV-infected *Rag2^{-/-}Il2rg^{-/-}* mice in comparison to WT mice, probably due to differences in the composition of the immune cell infiltrate between these two strains. Thus, the lack of all types of lymphocytes in *Rag2^{-/-}Il2rg^{-/-}* mice might influence the overall inflammatory status of these mice, and, therefore, the production of IL-12 by activated myeloid cells. Moreover, the lower amounts of $\text{Ifn}\gamma$ in *Rag2^{-/-}Il2rg^{-/-}* mice (only produced by eNK cells) might also affect the positive feedback loop of this cytokine on IL-12 production.

3. In the flow analyses in Figure 2, the authors should gate for the NK1.1⁻ subset of CD49b⁺CD122⁺ cells from WT and *Rag2^{-/-}Il2rg^{-/-}* mice and analyze more extensive marker expression including CD94 and Ly49 as they are important markers to differentiate immature and mature NK cells.

We agree. As requested, we have analyzed the expression of CD94, NKG2A/C/E and Ly49 receptors in the NK1.1^{low} subset of CD49b⁺CD122⁺ cells from IL-12-treated WT mice (NK1.1^{low} NK + IL-12) and compared it to that of NK1.1⁺CD49b⁺CD122⁺ NK cells from Ctrl- (NK) or IL-12-treated WT mice (NK + IL-12), as well as eNK cells from *Rag2^{-/-}Il2rg^{-/-}* mice. We found remarkable similarities between eNK cells and NK1.1^{low} NK + IL-12 cells from WT mice, such as high levels of CD94 and NKG2A/C/E, but a low expression of Ly49D and Ly49I when compared to NK or NK1.1⁺ NK + IL-12 cells from WT mice (See **new Fig. 3d**). The levels of Ly49G2 and Ly49A in NK1.1^{low} NK + IL-12 cells were, however, more similar to those from NK1.1⁺ NK + IL-12 cells (See **new Fig. 3d**). These results place eNK cells from *Rag2^{-/-}Il2rg^{-/-}* mice and NK1.1^{low} NK + IL-12 cells from WT mice phenotypically close to the immature population of CD122⁺NK1.1⁺CD94⁺Ly49⁻CD49b^{dim}CD11b^{dim} conventional NK cells (Vosshenrich *et al.*, *Curr Opin Immunol* 2005). Importantly, the CD94^{high} subset of NK cells was found to be hyperproliferative, showed an exclusive expression of NKG2A/C/E, high levels of CD69 but low expression of Ly49D and Ly49G2 (Yu *et al.*, *J Immunol* 2009), in accordance with the phenotype observed for eNK cells throughout our study (Fig. 2 and Fig. 3). The fact that both eNK cells and NK1.1^{low} NK + IL-12 cells express CD49b but only low levels of NK1.1 suggests, however, that IL-12 is able to generate a distinct cell subset with immature features, which further requires γ c-cytokines to develop into a fully mature population of NK cells.

4. In the transcriptome analyses (Figure 3), NK1.1^{low} cells from IL-12-injected WT mice should be purified and compared with those from *RAG2^{-/-}γc^{-/-}* mice. Because eNK-like cells in WT mice are rather rare, it is difficult to directly compare the same populations in WT and *RAG2^{-/-}γc^{-/-}* mice.

We agree with the reviewer on the importance to separate the NK1.1^{low} and NK1.1⁺ subset of CD49b⁺CD122⁺ cells induced by IL-12 in WT mice (NK + IL-12) to better compare them to eNK cell of *Rag2^{-/-}Il2rg^{-/-}* mice. However, instead of performing additional transcriptome analyses of these populations, we chose to analyze cell surface protein levels of the most differentially expressed genes by flow cytometry. In general, the phenotype of NK1.1^{low} NK + IL-12 cells from WT mice closely resembled that of eNK cells (See **new Fig. 2c-e, Fig. 3d, and Supplementary Fig. 1**). Thus, NK1.1^{low} NK + IL-12 cells showed a less differentiated and highly proliferative phenotype characterized with high levels of DNAM-1 and Eomes, as well as low levels of Nkp46 and Bcl-2 (See **new Fig. 2c-e and Supplementary Fig. 1**). Moreover, these cells expressed high levels of CD94, NKG2A/C/E and TRAIL, but low amounts of Ly49D and Ly49I when compared to NK or NK1.1⁺ NK + IL-12 cells from WT mice

(See **new Fig. 3d**). The resemblance between NK1.1^{low} NK + IL-12 cells from WT mice and eNK cells from mice again demonstrates the existence of an IL-12-induced pathway of alternative NK lymphopoiesis, which can take place in WT mice in the presence of γ c-cytokines. These results have been added to the revised version of the manuscript (See **new Fig. 2 c-e, Fig. 3d, and Supplementary Fig. 1**).

5. The authors state that eNK cells express Klr (p7). However, Figure 3 shows very low, if any, expression of most Klr genes. As discussed above, the expression of KLRs should be tested by flow cytometry.

This point was also raised by Reviewer #1. We have thus analyzed the cell surface levels of different KLRs in the population of eNK cells from *Rag2^{-/-}Il2rg^{-/-}* mice. Importantly, several KLRs such as Ly49A (*Klra1*), Ly49D (*Klra4*) and Ly49I (*Klra9*) were found to be expressed by eNK cells albeit at lower levels compared to conventional NK cells (See **new Fig. 3d**). These findings closely match the transcriptome data (Fig. 3 and Supplementary Table 1), which shows relatively low counts of Ly49A (*Klra1*) and Ly49D (*Klra4*) but solid expression of Ly49I (*Klra9*), Ly49G2 (*Klra7*) and Ly49C (*Klra3*) (Fig. 3 and Supplementary Table 1). The levels of LY49C could not be analyzed by flow cytometry due to the lack of specific antibodies for this receptor.

6. The differentiation of NKPs into eNK cells *in vitro* in Figure 5 is not very convincing. Figure 5f only shows upregulation of CD49b on NKPs (CD122+CD49b-) by IL-12 stimulation *in vitro*. Whether they acquire Nkp46, and NKG2D should also be tested.

As requested by the reviewer, we have performed additional *in vitro* differentiation assays isolating NKPs from WT and *Rag2^{-/-}Il2rg^{-/-}* mice. Upon stimulation with IL-12, NKPs differentiated into eNK cells that acquired NKG2D and expressed Nkp46 (See **new Fig. 5f**).

7. Anti-metastasis effects of IL-12 injection have been reported previously. Although the results presented in Figure 6 support the authors' contention that IL-12-induced eNK cells may play a major role, authors should test the effects of eNK cell depletion for more conclusive results. The data in Figure 6f are difficult to interpret without proper controls. Ideally *Ifng^{-/-} Rag^{-/-}Il2rg^{-/-}* mice should be tested.

The reviewer is of course correct in that *Ifng^{-/-}Rag2^{-/-}Il2rg^{-/-}* mice would have been useful to interpret the dependence of tumor rejection on IFN- γ -producing eNK cells. However combining these 6 alleles and that in Balb/c would take more than 1 1/2 years of breeding and congenics.

However, to address this point and to strengthen the notion that eNKs are potent tumor killers, we now present additional data using TRAIL-sensitive MC38 tumor cells (See **new Fig. 6a**). eNK cells were able to eliminate MC38 tumor cells *in vivo*, an effect which was abolished upon their depletion with anti-asialo GM1. Moreover, depletion of eNK cells in 4T1 tumor-bearing *Rag2^{-/-}Il2rg^{-/-}* mice

could reverse the inhibitory effects of IL-12 on lung metastasis (See **new Fig. 6h**), demonstrating the importance of this innate cell type in tumor surveillance.

8. Supplementary table 1 should include transcription factors known to be important for NK cell development, e.g. Nfil3 and Id2.

We agree. We have added this information in the updated version of the manuscript (See **new Supplementary Table 1**).

Reviewer #4, expert in NK biology (Remarks to the Author):

Ohs and colleagues identify an alternative pathway of NK cell development driven by the proinflammatory cytokine IL-12, which can occur independently of common gamma chain signaling. Administration of IL-12 or induction of this cytokine by virus infection was sufficient to elicit the emergence of a population of unique NK cells by specifically targeting NK cell precursors (NKPs) in the bone marrow (BM). These unique NK cells were able to limit metastatic growth in a mouse model of breast cancer. These data reveal a novel IL-12-driven hard-wired pathway of emergency NK lymphopoiesis bypassing steady state common gamma chain signaling from cytokines like IL-15. The findings are novel and advance the field by revealing potential therapeutic means of inducing NK cells in SCID, NK cell-deficient, or other patients.

We thank the reviewer for the astute summary and positive evaluation.

Major issues:

Too little data is provided to effectively distinguish these eNK from ILC1. Namely, the staining for CD127 expression that is described should be shown in the manuscript, and other defined markers that distinguish these cells should be explored if possible.

Indeed, eNK cells display an immature phenotype (See Fig. 3), and share many ILC1 markers (Robinette *et al.*, Nature Immunology 2015, Yu *et al.*, Trends Immunology 2013). We are fairly agnostic as to whether we should call them ILC1 or NK cells. However, as requested, we have added the missing information on IL-7 Receptor (CD127) expression (See **new Supplementary Fig. 1b**). We also analyzed levels of CD49a (proposed by Spits *et al.*, Nat Immunol. 2016 to be differentially expressed on ILC1s). Importantly, while eNK cells express some CD49a (see graph below), they not only lack CD127, but express high levels of Eomes (Fig. 2). Thus, according to the data published by Klose *et al.* in 2014, we classified our eNK cells as *bona fide* NK cells and not ILC1s.

Quantification of the % of CD49a in NK cells from Ctrl-treated WT mice (NK), the NK1.1⁺ and NK1.1^{low} subsets of CD122⁺CD49b⁺ NK cells from IL-12-treated WT mice (NK1.1⁺ NK + IL-12 and NK1.1^{low} NK + IL-12, respectively) and eNK cells from *Rag2*^{-/-}*Il2rg*^{-/-} mice. Each dot represents one mouse. **p* < 0.05, (ANOVA test with Bonferroni post-test).

The YAC-1 cell killing in Figure 2F is very unconvincing. The data shown is unclear with regards to how clean the staining with Topro was and whether YAC-1 cells were lost to cell death, the protocol for Topro staining and interpretation of results is poorly described, and the assay is performed only at low E:T ratios. There are more accurate means of measuring NK cell killing of YAC-1 cells than that used by the authors (i.e. chromium release).

The regulatory authorities no longer permit the use of chromium-51 in cellular killing assays. The staining with Topro on our tumor cells was quite crisp though, as shown with the spontaneous lysis of YAC cells (around 5%) in the absence of NK cells, which was used to quantify the percentage of specific lysis (Fig. 2f). However, to interrogate the cytotoxic properties of eNK cells we used the TRAIL-sensitive MC38 tumor cells as well as the 4T1 cells as targets *in vivo*. In both cases, eNK cells were able to eliminate tumor cells (See **new Fig. 6a,h**), as demonstrated through their depletion with an anti-asialo GM1 antibody.

How can the authors be certain that IL-12 in Figure 6 is not directly acting on the tumor?

We thank the reviewer for this comment. We now show that 4T1 tumor cells do not express *IL-12Rb1* and *IL-12Rb2* transcripts (See **new Supplementary Fig. 4a**). Moreover, IL-12 did not affect the growth/morphology of 4T1 cells *in vitro* (See **new Supplementary Fig. 4b**). These new results have been added to the revised version of the manuscript.

What evidence is there that the eNK cells are required for this anti-metastatic effect?

This is an important point and we investigated that in some more detail. As requested by the reviewer, the anti-tumor properties of eNK cells have been examined in more detail in the 4T1 model of lung metastasis. As shown in **new Fig. 6h**, eNK cell depletion using an anti-asialo GM1

antibody completely reversed the anti-metastatic effects of IL-12 in 4T1 tumor-bearing *Rag2^{-/-}Il2rg^{-/-}* mice (See **new Fig. 6h**), clearly demonstrating the importance of this innate cell type in the control of lung metastasis.

This reviewer was not clear from the experiments shown or the authors interpretation whether these eNK cells were entirely de novo generated following infection or administration of IL-12, or whether a small subset of these cells was simply expanded. Clearly the effect of IL-12 in vitro can drive differentiation of NKPs, but is this major mechanism in vivo. Maybe data from uninfected mice in Figure 1 demonstrating absence of these cells would be helpful. We agree, and, as suggested by the reviewer, we closely analyzed the presence of eNK cells in CD49b⁺CD122⁺ cells and found that the few events observed in PBS-treated *Rag2^{-/-}Il2rg^{-/-}* mice originated from potential myeloid cells that were not gated out properly (as discussed also above in response to Rev 1 and **new Fig. 1**). Therefore, we have included an additional step in our gating strategy and pregated on lymphocytes, resulting in the absence of CD49b⁺CD122⁺ cells in lungs of PBS-treated *Rag2^{-/-}Il2rg^{-/-}* mice (See **new Fig. 1a**). Also in Ctrl-treated *Rag2^{-/-}Il2rg^{-/-}* mice (Fig. 2) CD49b⁺CD122⁺ cells were absent from the lung. These data and the entire gating strategy has been corrected and incorporated into the new version of the manuscript.

At the top of page 5 and in other places, the conclusion that IL-12 is the main or only cytokine ("our findings exclude the requirement of other factors besides IL-12...", page11) driving this emergency lymphopoiesis is not rigorously tested since no other cytokines are examined, co-contributions from supporting cytokines are not addressed, and since blockade experiments do not distinguish between IL-12 (p40 homodimers or p70) and IL-23.

We agree that our findings do not exclude the role of other cytokines in NK cell emergency lymphopoiesis. However, our work definitely shows that IL-12 is sufficient to promote eNK cell development in the absence of an infection (Fig. 2) and therefore, has a key role (maybe not exclusive) in this process. We have reworded the language accordingly in the revised manuscript.

- Page 5: "These results highlight IL-12 as a key cytokine to initiate the emergence of γ c-signaling-independent CD122⁺CD49b⁺ cells..."

- Page 12: "Our findings show that IL-12 is sufficient for the generation of NK-like cells..."

Minor issues:

Experimental details are missing: 1) duration of VV infection prior to analysis, 2) which species of IL-12 is measured in Figure 1B, 3) what species of IL-12 is administered in vivo, and 4) how many experiments constitutes 'multiple' in the Figure 2A legend.

We agree and this information has been added in the revised version in the manuscript. The duration of VV infection has now been incorporated in the **new Figure 1c**. The species of IL-12 measured in Figure 1b is mouse, as stated in the Material and Methods section (see ELISA). The species of IL-12 administered *in vivo* is mouse and the details of its production is explained in a previous paper by vom Berg *et al.*, added in our reference list (vom Berg *et al.*, JEM 2013). The “multiple” experiments in Fig. 2a refer to 5 independent experiments. The figure legend has been changed accordingly.

Why was the presence of NK cells in the liver or other tissues not examined?

We thank the reviewer for this point. We have analyzed the presence and the phenotype of eNK cells in livers from WT and *Rag2^{-/-}Il2rg^{-/-}* mice and added these data in the revised version of the manuscript (See **new Supplementary Fig. 2**). Importantly, the population of eNK cells is also found in the liver of WT and *Rag2^{-/-}Il2rg^{-/-}* mice, and their phenotypic properties are very similar to the same cells found in lung tissue (See **new Supplementary Fig. 2 vs. Fig. 3**). These data has been incorporated to the revised version of the manuscript.

In addition to the representative data shown in Figure 2C, it would be useful to provide graphs revealing the mean and deviation of these various parameters (proportion and MFI of DNAM-1, etc.) across multiple mice.

We agree and the main differentially expressed markers on eNK cells (NKp46, DNAM-1, NKG2D, Eomes, IL7R, CD11b, CD27, KLRG1, Ki67 and Bcl-2) have been quantified accordingly and the results added in **new Fig. 2 and Supplementary Figure 1**.

The authors should stick with one naming convention and avoid the switch from eNK to NK-IL-12 in Figure 5.

We totally agree and the nomenclature in Fig. 5 has been changed to eNK cells to be consistent with the rest of the manuscript.

This reviewer finds no demonstration of unbridled protective properties in the data that would substantiate the conclusion drawn by the authors on page 6.

The sentence has been adjusted to the following: “Taken together, IL-12 bypasses the requirement of γ c-signaling for NK lymphopoiesis by inducing the generation of an unconventional population of eNK cells with cytotoxic properties”.

REVIEWERS' COMMENTS:

Reviewer #1 (Remarks to the Author):

I have read the detailed response to my original comments from the authors and I feel they have addressed each of these comments as thoroughly as can reasonably be asked of them. They have made the appropriate changes to the text and figures. I now recommend this paper be accepted for publication.

Reviewer #2 (Remarks to the Author):

The authors have satisfactorily addressed the points raised in the previous round of review.

Reviewer #3 (Remarks to the Author):

The authors have properly addressed the concerns raised to the previous version and revised the manuscript accordingly.

Reviewer #4 (Remarks to the Author):

The authors have done an excellent job addressing all the points made by the referees. The new data and revisions to the manuscript provide a stronger basis for the novel conclusions presented.